# REGULARIZED AUTOENCODERS FOR ISOMETRIC REPRESENTATION LEARNING

**Yonghyeon Lee[1], Sangwoong Yoon[1], Minjun Son[1], Frank C. Park[1,2]**
[1] Department of Mechanical Engineering, Seoul National University
[2] Saige Research
{yhlee,swyoon,mjson98}@robotics.snu.ac.kr, fcp@snu.ac.kr

## ABSTRACT

The recent success of autoencoders for representation learning can be traced in large part to the addition of a regularization term. Such regularized autoencoders "constrain" the representation so as to prevent overfitting to the data while producing a parsimonious generative model. A regularized autoencoder should in principle learn not only the data manifold, but also a set of geometry-preserving coordinates for the latent representation space; by geometry-preserving we mean that the latent space representation should attempt to preserve actual distances and angles on the data manifold. In this paper we first formulate a hierarchy for geometry-preserving mappings (isometry, conformal mapping of degree $k$, area-preserving mappings). We then show that a conformal regularization term of degree zero – i.e., one that attempts to preserve angles and relative distances, instead of angles and exact distances – produces data representations that are superior to other existing methods. Applying our algorithm to an unsupervised information retrieval task for CelebA data with 40 annotations, we achieve 79% precision at five retrieved images, an improvement of more than 10% compared to recent related work. Code is available at https://github.com/Gabe-YHLee/IRVAE-public.

## 1 INTRODUCTION

Learning a good representation for high-dimensional data is one of the most fundamental problems in machine learning. A good representation should capture all essential information about the data in a parsimonious manner while filtering out all non-essential variations. Clearly what is "essential" or "non-essential" depends heavily on the end task (Tschannen et al., 2018), and numerous criteria have been proposed for a range of contexts, e.g., disentanglement (Higgins et al., 2017; Zhao et al., 2017), clustering (Makhzani et al., 2015), sparsity (Lewicki & Sejnowski, 2000), hierarchy (Gulrajani et al., 2017), and isometric embedding (McQueen et al., 2016; Peterfreund et al., 2020).

In this paper we take the view that the essential information is best captured by the geometry of the data. More specifically, we adopt the manifold hypothesis as our point of departure, and further argue that a good representation should also preserve the geometry of the data manifold. That is, nearby points on the manifold should have representations in the latent representation space that are also nearby, and angles and volumes should be preserved as much as possible when moving between the data manifold and its representation space. To find such a representation, it is important to (i) learn the correct low-dimensional data manifold, and (ii) to find an optimal set of latent space coordinates that preserves the geometry of the learned data manifold.

A primary reason that autoencoders are widely used for unsupervised representation learning is that they can learn both the manifold and the latent space coordinates simultaneously during the training phase. Vanilla autoencoders trained purely to minimize reconstruction loss tend to overfit, and the learned manifolds are often inaccurate. As demonstrated in (Rifai et al., 2011b; Kingma & Welling, 2014; Makhzani et al., 2015; Tolstikhin et al., 2018; Lee et al., 2021), by augmenting the reconstruction loss with a regularization term, manifold learning performance of autoencoders can be significantly enhanced. However, little if any consideration has been given to the concurrent problem of learning a set of latent space coordinates that preserves the geometry of the data manifold (with the exception of FMVAE (Chen et al., 2020), which we discuss further below).

The main contribution of this paper is a regularized autoencoder that simultaneously learns the data manifold and a geometry-preserving set of latent space coordinates. For this purpose, we first formulate a hierarchy of geometry-preserving mappings. At the top of the hierarchy are isometries, which preserve distances and angles, followed by conformal maps, which preserve angles, and then area-preserving maps. Of particular relevance to this paper are the conformal maps, which we further stratify into degree $k$ conformal maps, with $k = 0$ corresponding to *scaled isometries*, i.e., maps that preserve angles and distances up to some scale factor.

Based on this hierarchy of mappings, we then formulate a corresponding hierarchy of regularization criteria for training the autoencoder. One of the important findings of our study is that somewhat counterintuitively, using a regularization term that measures the nearness to an isometry is in fact detrimental; such a regularization term overly constrains the mapping, resulting in a higher reconstruction loss whose effects cannot be mitigated even with adjustments to the regularization term weight. Rather, using a less stringent regularization term is more helpful. In our examples the degree zero conformal maps, or scaled isometries, seem to offer the best balance between reconstruction accuracy and model parsimony.

We note that if the exact data manifold were known in advance, then the scale factor could be pre-computed, e.g., to make the latent space and the data manifold have the same volume, in which case it would make sense to use a normalized version of the isometry measure as regularization term. When using an autoencoder for representation learning, however, the data manifold is not known a priori but rather learned together with the latent space representation. The more effective alternative, we argue, is to learn this scale together with the manifold and the latent space representation. The FMVAE of (Chen et al., 2020) is in fact the first work to adopt this approach, but the measure does not adequately capture the nearness to a scaled isometry, and is also not coordinate-invariant, limiting its performance as we show later in our experiments.

Once the data manifold and an initial set of geometry-preserving latent space coordinates are learned, it is possible to further "flatten" the latent space, by adding a postprocessing step that leads to an even more isometric set of coordinates. Specifically, we use an invertible neural network model to map the pre-trained latent space to a more isometric representation space, without incurring any further losses in reconstruction accuracy.

Experiments on diverse image and motion capture data confirm that, compared to existing related methods, our geometrically regularized autoencoder produces more isometric representations of the data while incurring only minimal losses in reconstruction accuracy. In particular, information retrieval task experiments conducted with *CelebA* data show that data similarity measurements performed in our representation space lead to significantly improved levels of retrieval performance.

Our specific contributions can be summarized as follows:

- We define a family of coordinate-invariant regularization terms that measure how close the decoder is to being a scaled isometry;
- We propose an isometric regularization method for autoencoders that learns both the data manifold and a set of geometry-preserving latent space coordinates, all while incurring minimal losses in reconstruction accuracy;
- We propose a postprocessing flattening technique that learns a more isometric representation space without further losses in reconstruction accuracy.

## 2 A HIERARCHY OF GEOMETRY-PRESERVING MAPPINGS

This section introduces a hierarchy of geometry-preserving mappings between two Riemannian manifolds. Let $\mathcal{M}$ be a Riemannian manifold of dimension $m$ with local coordinates $z \in \mathbb{R}^m$ and Riemannian metric $G(z) \in \mathbb{R}^{m \times m}$, and $\mathcal{N}$ be a Riemannian manifold of dimension $n$ with local coordinates $x \in \mathbb{R}^n$ and Riemannian metric $H(x) \in \mathbb{R}^{n \times n}$. Let $f : \mathcal{M} \to \mathcal{N}$ be a smooth mapping, represented in local coordinates by the italic symbol $f : \mathbb{R}^m \to \mathbb{R}^n$. Its differential is denoted by the Jacobian matrix $J_f(z) := \frac{\partial f}{\partial z}(z) \in \mathbb{R}^{n \times m}$.

Intuitively, an *isometry* is a mapping between two spaces that preserves distances and angles everywhere. For a linear mapping between two vector spaces equipped with inner products, an isometry preserves the inner product everywhere. In the case of a mapping between Riemannian manifolds,

$f : \mathcal{M} \to \mathcal{N}$ is an isometry if

$$G(z) = J_f(z)^T H(f(z)) J_f(z) \quad \forall z \in \mathbb{R}^m. \tag{1}$$

Sometimes, requiring a map $f$ to be an isometry can be overly restrictive; preserving only angles may be sufficient. A *conformal map* is a mapping that preserves angles but not necessarily distances. Mathematically, $f : \mathcal{M} \to \mathcal{N}$ is conformal (or angle-preserving) if

$$G(z) = c(z) J_f(z)^T H(f(z)) J_f(z) \quad \forall z \in \mathbb{R}^m, \tag{2}$$

for some positive function $c : \mathcal{M} \to \mathbb{R}$. The positive function is called the conformal factor.

Conformal maps can be further categorized by the polynomial degree of $c(z)$. A conformal map of degree zero, i.e., one in which $c(z)$ is constant, sits one level below the isometric mapping and is defined formally as any mapping $f$ for which a positive scalar constant $c$ satisfying

$$G(z) = c J_f(z)^T H(f(z)) J_f(z) \quad \forall z \in \mathbb{R}^m \tag{3}$$

can be found. Such a map not only preserves angles but also scaled distances; for this reason we shall also refer to a degree zero conformal map as a *scaled isometry*. Area-preserving maps also can be placed within this hierarchy, but for the purposes of this paper our focus will be exclusively on isometric and conformal mappings.

## 3 A COORDINATE-INVARIANT RELAXED DISTORTION MEASURE

The goal of this section is to design a coordinate-invariant functional $\mathcal{F}$ that measures the proximity of the mapping $f : \mathcal{M} \to \mathcal{N}$ to a scaled isometry. Section 3.1 shows how to construct coordinate-invariant functionals for a smooth mapping between two Riemannian manifolds, while Section 3.2 introduces a simple technique to define coordinate-invariant distortion measures that measure how close the mapping is to being an isometry. Section 3.3 defines a family of coordinate-invariant *relaxed* distortion measures that measure how close the mapping is to being a scaled isometry.

In order to extend the discussion of traditional distortion measures to more general cases (we will later use a probability measure), we consider a positive measure $\nu$ on $\mathcal{M}$ absolutely continuous to the Riemannian volume form[1], the spaces of interest will then be limited to the support of $\nu$ rather than the entire manifold $\mathcal{M}$.

### 3.1 COORDINATE-INVARIANT FUNCTIONALS ON RIEMANNIAN MANIFOLDS

We begin this section by reviewing how to construct coordinate-invariant functionals for a smooth mapping $f : \mathcal{M} \to \mathcal{N}$ (Jang et al., 2020). At a point $z \in \mathcal{M}$, consider the characteristic values of the pullback metric $J_f^T H J_f$ relative to the metric $G$ of $\mathcal{M}$, i.e., the $m$ eigenvalues $\lambda_1(z), ..., \lambda_m(z)$ of $J_f(z)^T H(f(z)) J_f(z)^T G^{-1}(z)$. Apart from their order, these eigenvalues are intrinsically associated with $J_f^T H J_f$ and $G$, i.e., they are invariant under coordinate transformations[2].

Let $S(\lambda_1, ..., \lambda_m)$ be any symmetric function (i.e., a function whose value is invariant with respect to permutations of its arguments) of the $m$ eigenvalues. Then the integral

$$\mathcal{I}_S(f) := \int_{\mathcal{M}} S(\lambda_1(z), ..., \lambda_m(z)) \, d\nu(z) \tag{4}$$

is an intrinsic quantity, i.e., coordinate-invariant.

In this paper we introduce a new family of coordinate-invariant functionals that can be used to formulate measures for scaled isometries in a more natural way. Consider a symmetric function $S$ such that the above integral $\mathcal{I}_S(f) \neq 0$, and let $S'(\lambda_1, ..., \lambda_m)$ be any symmetric function. The following integral

$$\int_{\mathcal{M}} S'(\frac{\lambda_1(z)}{\mathcal{I}_S(f)}, ..., \frac{\lambda_m(z)}{\mathcal{I}_S(f)}) \, d\nu(z) \tag{5}$$

is then an intrinsic quantity, since the eigenvalues and $\mathcal{I}_S(f)$ are both intrinsic quantities.

---

[1] Given a measurable space $\mathcal{M}$, a measure $\nu$ is absolutely continuous to the Riemannian volume measure Vol such that $d\text{Vol}(z) = \sqrt{\det G(z)} \, dz$, if $\text{Vol}(A) = 0$ implies $\nu(A) = 0$ for any measurable set $A \subset \mathcal{M}$.

[2] Eigenvalues of $J_f^T H J_f G^{-1}$ remain the same under coordinate transformations $z \mapsto \phi(z), x \mapsto \psi(x)$.

## 3.2 Distortion Measures of Isometry

Recall that f : $\mathcal{M} \to \mathcal{N}$ is a local isometry at $z$ if $\lambda_i(z) = 1$ for $\forall i$, and a global isometry if f is a local isometry everywhere (Jang et al., 2020). In this section we introduce a simple technique to define a family of distortion measures that measure the proximity to an isometry. Consider any convex function $h : \mathbb{R} \to [0, \infty)$ such that $h(1) = 0$, $h'(\lambda) = 0$ iff $\lambda = 1$, and define a symmetric function $S(\lambda_1, ..., \lambda_m) = \sum_{i=1}^m h(\lambda_i)$. Then the coordinate-invariant functional

$$\int_{\mathcal{M}} \sum_{i=1}^m h(\lambda_i(z)) \, d\nu(z) \tag{6}$$

is a global measure of distortion (restricted to the support of $\nu$). Since the integrand in the above functional locally measures the deviation of the mapping f from an isometry, its integral also serves as a global measure of distortion. Popular choices include $h(\lambda) = (1 - \lambda)^2$ and $h(\lambda) = (\log(\lambda))^2$.

## 3.3 A Relaxed Distortion Measure and Scaled Isometry

The goal of this section is to design a family of coordinate-invariant functionals $\mathcal{F}$ that measure how far the mapping f : $\mathcal{M} \to \mathcal{N}$ is from being a scaled isometry (over the support of $\nu$). We refer to these as *relaxed distortion measures*, in the sense that a larger set of mappings are minimizers of the relaxed distortion measure than the original distortion measure.

Given mappings f, f' : $\mathcal{M} \to \mathcal{N}$ (denoted $f$ and $f'$ in local coordinates) with respective Jacobians $J_f$, $J_{f'}$, the desired properties of $\mathcal{F}$ are as follows:

1. $\mathcal{F}(f) \geq 0$;
2. $\mathcal{F}(f) = 0$ if and only if $\lambda_i(z) = c$ for $\forall i$, $\forall z \in \text{Supp}(\nu)$, and for some $c > 0$;
3. $\mathcal{F}(f) = \mathcal{F}(f')$ if $J_f^T H J_f = c J_{f'}^T H J_{f'}$ for $\forall z \in \text{Supp}(\nu)$, and for some $c > 0$,

where $\lambda_i(z)$ denotes the eigenvalues of $J_f^T(z) H(f(z)) J_f(z) G^{-1}(z)$ and $\text{Supp}(\nu)$ is the support of $\nu$. The first condition is to ensure a minimum of zero, while the second condition is to make any scaled isometry (restricted to the support of $\nu$) be a minimizer. Although the first and second conditions are sufficient to use $\mathcal{F}(f)$ as a measure of the proximity of $f$ to a scaled isometry, we impose an additional third condition to make the measure more natural in the following sense: since the measure should not a priori favor a particular scale for the pullback metric, if the pullback metrics are equivalent up to some scale, then these should be treated equivalently.

Our core idea for constructing such a measure is to use the newly proposed family of coordinate-invariant functionals (5) with the technique used in (6) as follows:

$$\mathcal{F}(f) := \int_{\mathcal{M}} \sum_{i=1}^m h\left(\frac{\lambda_i(z)}{\int_{\mathcal{M}} S(\lambda_1(z), ..., \lambda_m(z)) \, d\nu(z)}\right) d\nu(z) \tag{7}$$

where $h$ is some convex function and $S$ is some symmetric function. This functional automatically satisfies the first condition. To satisfy the second and third conditions, the symmetric function $S$ must satisfy some further conditions:

**Proposition 1** *The functional in (7) satisfies the second and third conditions if $\nu$ is a finite measure, $S(k\lambda_1, ..., k\lambda_m) = kS(\lambda_1, ..., \lambda_m)$, and $S(1, ..., 1) = 1/\nu(\mathcal{M})$. Proof: See Appendix A.*

# 4 Isometric Representation Learning

We now introduce a regularized autoencoder with a practical form of the relaxed coordinate-invariant distortion measure. We also describe the additional postprocessing step for further flattening the latent space. Finally, we introduce some relevant implementation details.

## 4.1 Isometric Regularization with the Relaxed Distortion Measure

Denote a parametric encoder function by $g_\phi : \mathbb{R}^D \to \mathbb{R}^m$ and decoder function by $f_\theta : \mathbb{R}^m \to \mathbb{R}^D$. We consider the latent space $\mathbb{R}^m$ with coordinates $z$ assigned with the identity metric $G(z) = I$ and

the data space $\mathbb{R}^D$ with coordinates $x$ assigned with the metric $H(x)$. We denote the data distribution in $\mathbb{R}^D$ by $x \sim P_D$ and the distribution of the encoded data in $\mathbb{R}^m$ by $z \sim P_\phi$ where $z = g_\phi(x)$.

First, to construct a relaxed distortion measure of $f_\theta$, we need to select a positive measure $\nu$. Considering $\nu$ as a probability measure $P_\phi$ and replacing the integrals $\int_{\mathcal{M}} d\nu(z)$ in (7) by expectations $\mathbb{E}_{z \sim P_\phi}$, we get the following expression:

$$\mathcal{F}(f_\theta; P_\phi) := \mathbb{E}_{z \sim P_\phi}[\sum_{i=1}^{m} h(\frac{\lambda_i(z)}{\mathbb{E}_{z \sim P_\phi}[S(\lambda_1(z), ..., \lambda_m(z))]})], \tag{8}$$

where $\lambda_i(z)$ are the eigenvalues of the pullback metric $J_{f_\theta}^T(z) H(f_\theta(z)) J_{f_\theta}(z)$.

Then, we propose a regularized autoencoder where the loss function consists of the following two terms i) loss function $\mathcal{L}(\theta, \phi)$ for manifold learning (e.g., reconstruction error) and ii) regularization term $\mathcal{F}(f_\theta; P_\phi)$ for learning a scaled isometric decoder:

$$\min_{\theta, \phi} \mathcal{L}(\theta, \phi) + \alpha \mathcal{F}(f_\theta; P_\phi). \tag{9}$$

Training an autoencoder to minimize (9) is referred to as **Isometric Regularization (IR)**.

In practice, the computation of and back-propagation through $\mathcal{F}(f_\theta; P_\phi)$ that includes the computation of the entire Jacobian of $f_\theta$ requires sufficient memory and computational cost. Instead, with an appropriate choice of convex function $h$ and symmetric function $S$, the measure $\mathcal{F}(f_\theta; P_\phi)$ can be efficiently estimated by using the more easily computed Jacobian-vector and vector-Jacobian products and Hutchinson's stochastic trace estimator (Hutchinson, 1989):

**Proposition 2** *Let $h(\lambda) = (1 - \lambda)^2$ and $S(\lambda_1, ..., \lambda_m) = \sum_{i=1}^{m} \lambda_i/m$, then*

$$\mathcal{F}(f_\theta; P_\phi) = \mathbb{E}_{z \sim P_\phi}[\sum_{i=1}^{m}(\frac{\lambda_i(z)}{\mathbb{E}_{z \sim P_\phi}[\sum_i \lambda_i(z)/m]} - 1)^2] = m^2 \frac{\mathbb{E}_{z \sim P_\phi}[\text{Tr}(H_\theta^2(z))]}{\mathbb{E}_{z \sim P_\phi}[\text{Tr}(H_\theta(z))]^2} - m, \tag{10}$$

*where $\lambda_i(z)$ means the eigenvalues of $H_\theta(z) := J_{f_\theta}^T(z) H(f_\theta(z))) J_{f_\theta}(z)$. Proof: See Appendix A.*

## 4.2 Latent Space Flattening

In the isometric regularization approaches, autoencoders are trained to minimize both manifold learning loss and geometric regularization term; hence there exists an inherent tradeoff. We introduce a postprocessing step to further flatten the pre-trained latent space to a more isometric representation space without incurring any further losses in reconstruction accuracy.

Given a trained encoder $g_\phi : \mathbb{R}^D \to \mathbb{R}^m$ and trained decoder $f_\theta : \mathbb{R}^m \to \mathbb{R}^D$, consider an invertible map $i : \mathbb{R}^m \to \mathbb{R}^m$ such that $z' = i(z)$ and the composition $f_\theta \circ i^{-1} : \mathbb{R}^m \to \mathbb{R}^D$. The map $i$ transforms the pre-trained latent space to a new set of latent coordinates without affecting the reconstruction accuracy. This gives an additional degree of freedom to find a more isometric representation space. Based on this idea, we formulate the latent space flattening problem as follows:

$$\min_{i} \mathcal{F}(f_\theta \circ i^{-1}; P_i') + \beta \mathbb{E}_{z \sim P_\phi}[\|i(z)\|^2], \tag{11}$$

where $z' \sim P_i'$ is the distribution of the encoded data in $\mathbb{R}^m$ ($z' = i \circ g_\phi(x)$) and the second term is the regularization term with coefficient $\beta$ added to prevent $i(z)$ from diverging. In this paper, we use the invertible deep neural network *RealNVP* (Dinh et al., 2017) for $i$.

## 4.3 Implementation Details

We now explain some relevant implementation details of our approach. We use a Gaussian encoder $q_\phi(z|x)$ and decoder $p_\theta(x|z)$, and treat the mean of $p_\theta(x|z)$ as $f_\theta(z)$. We use the negative evidence lower bound for $\mathcal{L}(\theta, \phi)$ from the VAE with the unit Gaussian prior distribution (Kingma & Welling, 2014). We assume the identity metric for the data space, i.e., $H(x) = I$. We use the particular combination of $h$ and $S$ described in Proposition 2. In cases when the latent space dimension is sufficiently small, it becomes feasible to compute the full metric and use other combinations. As an example, when the dimension is two, we use another popular choice $h(\lambda) = (\log(\lambda))^2$.

**Augmented Distribution:** The relaxed distortion measure is defined as the expectation over the encoded data distribution $P$. However, the influence of the measure is then limited to regions where data is available; in practice, data augmentation can resolve this issue. Following the FMVAE (Chen et al., 2020), we use the modified mix-up data-augmentation method with $\eta > 0$, where $P$ is augmented by $z = \delta z_1 + (1 - \delta)z_2$ such that $z_i \sim P, i = 1, 2$, where $\delta$ is uniformly sampled from $[-\eta, 1 + \eta]$. Sampling from the augmented latent space data distribution is denoted as $z \sim P_Z$.

The pseudocode is available in Appendix B.

## 5   EXPERIMENTS

Throughout this section we use isometric regularization on VAE (Kingma & Welling, 2014; Rezende & Mohamed, 2015) which we refer to as the *Isometrically Regularized VAE (IRVAE)*. We then train the *Flattening Module (FM)* in a post hoc manner, which we denote by *IRVAE + FM*. In Section 5.1, we show that (i) IRVAE can effectively learn the isometric representation with a minimal loss in reconstruction accuracy compared to vanilla VAE and FMVAE (Chen et al., 2020), and (ii) IRVAE + FM leads to a more isometric representation without any loss in reconstruction accuracy. In Section 5.2, through an unsupervised human face retrieval task, we show that measuring data similarity in our isometric representation spaces significantly improves retrieval performance. A mathematical comparison of IRVAE with FMVAE is provided in Appendix C, with experimental details given in Appendix D.

### 5.1   ISOMETRIC REPRESENTATION WITH MINIMAL LOSS IN RECONSTRUCTION ACCURACY

We first introduce some evaluation metrics that measure (i) the accuracy of the learned manifolds, and (ii) how isometric the latent space is (i.e., how close the pullback metric $M(z) := J_f^T(z)J_f(z)$ is to $\{cI | c \in (0, \infty)\}$). These metrics are computed over the test datasets.

The accuracy of the learned manifolds is measured in terms of the mean square reconstruction errors. To evaluate $M(z)$, we introduce two different metrics that are scale-invariant (i.e., $M(z)$ and $kM(z)$ have the same values). First, the *Variance of the Riemannian metric (VoR)* is defined as the mean square distance from $M(z)$ to $\bar{M} := \mathbb{E}_{z \sim P_Z}[M(z)]$, where we use the affine-invariant Riemannian distance $d$, i.e., $d^2(A, B) = \sum_{i=1}^m (\log \lambda_i(B^{-1}A))^2$ (Pennec et al., 2006; Lee & Park, 2018). Second, we use the *Mean Condition Number (MCN)* of the Riemannian metric, where the condition number of $M(z)$ is the ratio between the maximum eigenvalue and minimum eigenvalue. Both metrics are computed by sampling a sufficiently large number of points from $P_Z$.

VoR measures how the Riemannian metric is spread out from $\bar{M}$, which becomes zero when the metric is constant for all $z$ in the support of $P_Z$. While VoR measures the spatial variance of the metric in $z$, MCN measures how much the metric is isotropic. MCN becomes one when $M(z) \propto I$ for all $z$ in the support of $P_Z$.

For IRVAE and FMVAE, there are natural tradeoffs between MSE versus VoR and MCN. If we use higher weights $\alpha$ for the regularization terms, the representation becomes more isometric with losses in reconstruction accuracy. In order to compare the algorithms in a regularization coefficient $\alpha$-invariant manner, we compare the tradeoff curves, i.e., MSE as a function of VoR and MCN obtained by using varying regularization coefficients $\alpha$.

Sections 5.1.1 and 5.1.2 use the *MNIST* image and *CMU motion capture* data, respectively, with two-dimensional latent spaces. Additional experimental results including experiments on more diverse image data (*MNIST, FMNIST, SVHN, CIFAR-10*) are provided in Appendix E.

### 5.1.1   MNIST

Figure 1 shows tradeoff curves of FMVAE and IRVAE trained on MNIST data with two-dimensional latent spaces, some example reconstructed images, and latent space representations with equidistance plots. We note that the tradeoff curves of IRVAE (orange) are below those of FMVAE (blue), meaning that the IRVAE learns more isometric representations at any level of reconstruction accuracy. Cases A and B produce good reconstruction results, but non-homogeneous and non-isotropic

equidistance plots since lower regularization coefficients are used. For cases F, I, opposite results are obtained.

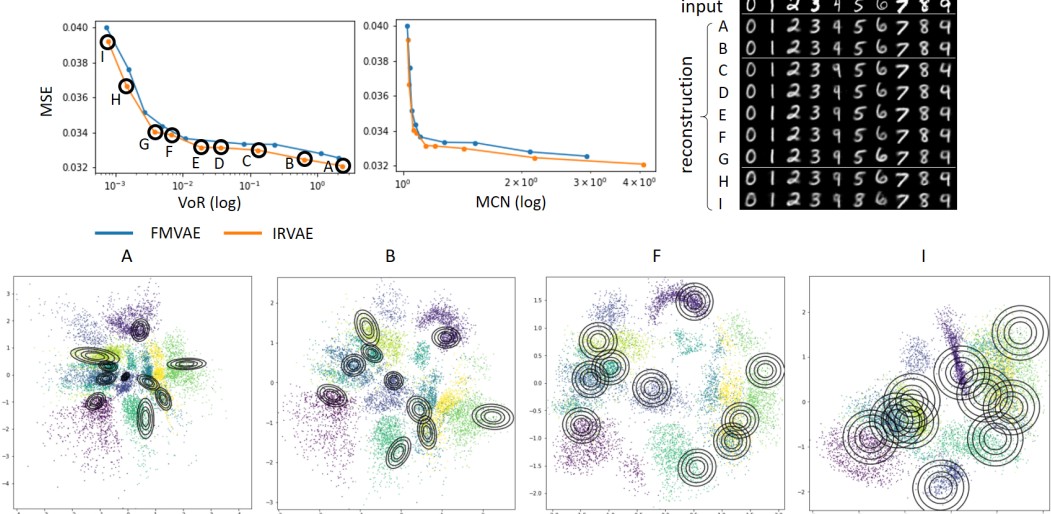

Figure 1: *Top:* The MSE and VoR, MCN tradeoff curves, and some example reconstructed images produced by IRVAE trained with various regularization coefficients. *Bottom:* Two-dimensional latent space representations with some equidistance plots whose centers consist of a randomly selected data point $z_c$ from each class for A, B, F, I. The equidistance plots are $\{z|(z-z_c)^T J_f^T(z_c) J_f(z_c)(z-z_c) = k$ for $k > 0$. (The more homogeneous and isotropic, the better.)

Figure 2 shows tradeoff curves and latent space representations for IRVAE + FM (under the same experimental setting as above). IRVAE + FM results in more isometric representations than IRVAE, with no losses in MSE. In particular, for cases A and B, which have the lowest MSE but the highest VoR and MCN, the flattener significantly lowers both VoR and MCN. These improvements can be qualitatively seen by comparing the equidistance plots of A and B in Figure 1 and Figure 2.

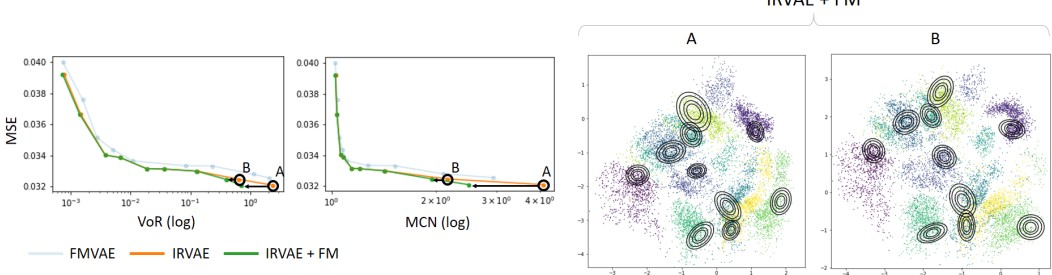

Figure 2: Tradeoff curves for FMVAE, IRVAE, and IRVAE + FM, and two-dimensional latent space representations with some equidistance plots (under the same experimental setting as Figure 1).

### 5.1.2 CMU MOTION CAPTURE DATA

We use a subset of the CMU motion capture data by selecting four classes: walking, jogging, balancing, and punching, with two-dimensional latent spaces. The motion data in each class is a sequence of pose data, where each pose data is represented by a 50-dimensional vector of joint angles. We use a total of 10,000, 2,000 and 2,000 pose data items for training, validation, and test, respectively.

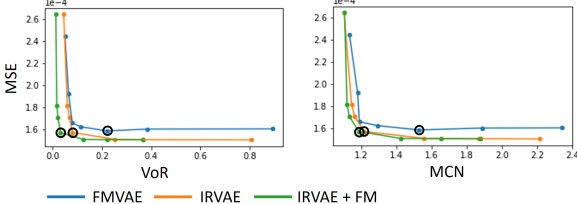

Figure 3: The tradeoff curves for FMVAE, IRVAE, IRVAE + FM trained with the pose data.

Figure 3 shows the tradeoff curves for FMVAE, IRVAE, and IRVAE + FM. The results of VAE are omitted from the figure since the MSE, VoR, and MCN are too large to be plotted together with the other models (MSE: $1.72 \times 10^{-4}$, VoR: 4.86, MCN: 20.3). The tradeoff curves for IRVAE are far below compared to those for FMVAE, while the IRVAE + FM results in a more isometric representation without any loss in MSE.

Figure 4 shows the latent space representations with equidistance ellipses. For a fair comparison, we select models with similar reconstruction accuracy (the selected regularization coefficients are marked by the black circles in Figure 3). We observe that the ellipses for IRVAE and IRVAE + FM are much more homogeneous (blue) than those for VAE and FMVAE.

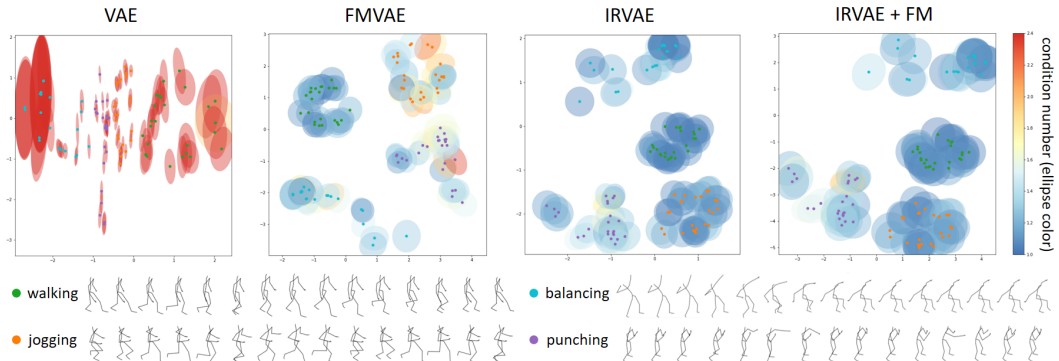

Figure 4: Latent spaces and equidistance ellipses for VAE, FMVAE, IRVAE, and IRVAE + FM (the redder the ellipse, the larger the condition number).

## 5.2 Unsupervised Human Face Retrieval

We also demonstrate the effectiveness of our approach to image-to-image retrieval of human faces. We consider the retrieval of face images that have a queried set of attributes, in which the query is also given as a set of face images that share some visual attributes. For example, if a user provides a set of photos that contain smiling faces but differ in other attributes, a retrieval algorithm is expected to return a set of smiling face images from the database. In this setting, a user communicates with a retrieval system by providing examples, avoiding the dependence on human language.

We focus in particular on an unsupervised scenario in which a retrieval algorithm is built without any annotation. Unsupervised representation learning methods are trained on CelebA (Liu et al., 2015), which contains 182,637 training images and 19,962 test images. Each image in CelebA has $64 \times 64 \times 3$ pixels and contains an aligned human face with binary annotations on 40 attributes. The attribute annotations are only used to evaluate retrieval performance.

We refer to attributes that query images have in common as query attributes. We experiment with different settings in which there is a single query attribute (single attribute retrieval; SAR) and also with two query attributes (double attribute retrieval; DAR). We use all images that have given query attributes in the training set as query images, and retrieve the top K similar images from the test set based on the summed cosine similarity to the query images.

The quality of a retrieval is measured using precision at K (P@K) for $K = 1, 5, 10, 20$. P@K denotes the ratio of images that have query attributes among the K retrieved images. For SAR, P@K are averaged over the 40 attributes. For DAR, there are 780 possible combinations of two attributes. We select a combination of attributes if the images that have those attributes account for more than one percent and less than fifty percent of the total test data; thus P@K are averaged over 487 selected combinations.

Retrieval is performed in the latent space of the representation learning algorithms. Algorithms to be compared include (i) unsupervised representation learning methods VAE (Kingma & Welling, 2014), FMVAE (Chen et al., 2020), IRVAE (ours), and IRVAE + FM (ours), whose latent space dimensions are 128, (ii) a neural network (ResNet-50) pre-trained on ImageNet (He et al., 2016), and (iii) a supervised learning method (binary relevance) (Read et al., 2011). The performance of

the pre-trained network and the supervised learning method serve as lower and upper bounds for the unsupervised methods, respectively.

Table 1: Precision at K retrieved images, i.e., P@K, averaged over the attributes. The best results among unsupervised methods are colored red, while the second best results are colored blue.

| Method | P@1 | P@5 | P@10 | P@20 | P@1 | P@5 | P@10 | P@20 |
|---|---|---|---|---|---|---|---|---|
| | *Single Attribute Retrieval* | | | | *Double Attribute Retrieval* | | | |
| VAE | 60.0 | 60.5 | 61.0 | 58.4 | 39.8 | 38.7 | 37.0 | 34.7 |
| FMVAE | 67.5 | 66.0 | 64.0 | 62.6 | 41.5 | 39.1 | 38.8 | 36.8 |
| IRVAE (ours) | **82.5** | 72.0 | 71.5 | 69.0 | 56.1 | 50.8 | 47.2 | 44.2 |
| IRVAE + FM (ours) | 75.0 | **79.0** | **75.3** | **74.1** | **57.3** | **54.0** | **51.4** | **49.4** |
| Pre-trained (ResNet-50) | 35.0 | 28.5 | 23.5 | 23.9 | 18.3 | 11.0 | 8.7 | 9.2 |
| Supervised (BR) | 87.5 | 82.5 | 80.3 | 80.8 | 66.9 | 65.2 | 63.7 | 62.6 |

Table 1 lists the retrieval performance for the various cases. IRVAE outperforms FMVAE, which in turn outperforms VAE. FM improves the performance of IRVAE for most cases with only one exception, P@1 score in SAR. More detailed results on SAR are included in Appendix F. Some example image retrieval results are visualized in Figure 5.

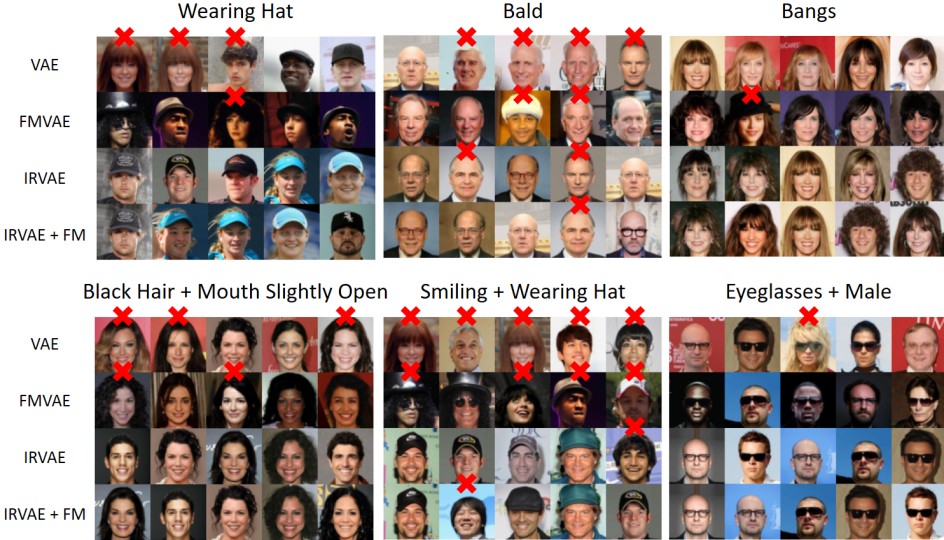

Figure 5: Some example image retrieval results (top 5 images). Common attributes of query image sets are written above the figures. Higher rank images are located left.

## 6 CONCLUSION

We have formulated the problem of learning the manifold simultaneously with a set of optimal latent space coordinates that preserve the geometry of the learned manifold. We have introduced a hierarchy of geometry-preserving mappings between two Riemannian manifolds (e.g., isometry, conformal mapping of degree $k$, area-preserving mappings) and defined a family of coordinate-invariant relaxed distortion measures that measure the proximity of the mapping to a scaled isometry (i.e., conformal mapping of degree 0). Finally, two algorithms, isometric regularization and latent space flattening, have been proposed. We have verified the efficacy of our methods with diverse image and motion capture data, and through a human face retrieval task with CelebA data.

We believe the algorithm can be further enhanced in a number of different ways. The current implementation of IRVAE and IRVAE + FM assumes the identity metric in the ambient data space, i.e., $H(x) = I$. Although this is a reasonable choice in a fully unsupervised setting, we think that domain-specific knowledge or a few labels can be leveraged to define better $H(x)$. In addition, instead of the mix-up data augmentation method used in this paper, developing a principled approach for defining $P_Z$ will be an interesting future research direction.

REPRODUCIBILITY STATEMENT

We have included complete proofs of the Propositions in Appendix A. Also, we have included experiment settings in detail as much as possible in Appendix D and E such as the number of training/validation/test splits of datasets, preprocessing methods, neural network architectures, and hyper-parameters used in model training (e.g., batch size, number of epochs, learning rate, etc).

ACKNOWLEDGMENTS

This work was supported in part by SRRC NRF grant 2016R1A5A1938472, IITP-MSIT Grant 2021-0-02068 (SNU AI Innovation Hub), SNU-IAMD, SNU BK21+ Program in Mechanical Engineering, and the SNU Institute for Engineering Research.

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

# APPENDIX

## A PROOF OF PROPOSITIONS

**Proof 1 (Proof of Proposition 1)** *For sufficiency* $(\Rightarrow)$ *of the second condition, suppose* $\mathcal{F}(f) = 0$. *Then, denoting by* $c = \int_{\mathcal{M}} S(\lambda_1(z), ..., \lambda_m(z)) \, d\nu(z)$,

$$\sum_{i=1}^{m} h(\frac{\lambda_i(z)}{c}) = 0 \quad \forall z \in \mathrm{Supp}(\nu).$$

*Since* $h(\lambda) \geq 0$ *for* $\forall \lambda \in \mathbb{R}$, $h(\lambda_i(z)/c) = 0$ *for* $\forall i, \forall z \in \mathrm{Supp}(\nu)$. *Since* $h'(\lambda) = 0$ *iff* $\lambda = 1$, $\lambda_i(z) = c$ *for* $\forall i, \forall z \in \mathrm{Supp}(\nu)$.

*For necessity* $(\Leftarrow)$ *of the second condition, suppose* $\lambda_i(z) = c$ *for* $\forall i, \forall z \in \mathrm{Supp}(\nu)$, *then*

$$\frac{\lambda_i(z)}{\int_{\mathcal{M}} S(\lambda_1(z), ..., \lambda_m(z)) \, d\nu(z)} = \frac{c}{\int_{\mathcal{M}} S(c, ..., c) \, d\nu(z)}$$

$$= \frac{c}{c \int_{\mathcal{M}} S(1, ..., 1) \, d\nu(z)}$$

$$= \frac{1}{\frac{1}{\nu(\mathcal{M})} \int_{\mathcal{M}} d\nu(z)}$$

$$= 1 \, for \, \forall z \in \mathrm{Supp}(\nu).$$

*Then consequently,*

$$\mathcal{F}(f) = \int_{\mathcal{M}} \sum_{i=1}^{m} h(\frac{\lambda_i(z)}{\int_{\mathcal{M}} S(\lambda_1(z), ..., \lambda_m(z)) \, d\nu(z)}) \, d\nu(z) = \int_{\mathcal{M}} \sum_{i=1}^{m} h(1) \, d\nu(z) = 0.$$

*For the third condition, suppose* $J_f^T H J_f = c J_g^T H J_g$ *in* $\mathrm{Supp}(\nu)$ *for some* $c > 0$. *Denote the eigenvalues of* $J_f^T H J_f G^{-1}$ *and* $J_g^T H J_g G^{-1}$ *by* $\lambda_i[f]$ *and* $\lambda_i[g]$ *where* $\lambda_1[f] \geq ... \geq \lambda_m[f]$ *and* $\lambda_1[g] \geq ... \geq \lambda_m[g]$, *respectively. Then, obviously,* $\lambda_i[f] = c\lambda_i[g]$ *for* $\forall i$. *Then consequently,*

$$\mathcal{F}(f) = \int_{\mathcal{M}} \sum_{i=1}^{m} h(\frac{\lambda_i[f](z)}{\int_{\mathcal{M}} S(\lambda_1[f](z), ..., \lambda_m[f](z)) \, d\nu(z)}) \, d\nu(z)$$

$$= \int_{\mathcal{M}} \sum_{i=1}^{m} h(\frac{c\lambda_i[g](z)}{\int_{\mathcal{M}} c \cdot S(\lambda_1[g](z), ..., \lambda_m[g](z)) \, d\nu(z)}) \, d\nu(z)$$

$$= \int_{\mathcal{M}} \sum_{i=1}^{m} h(\frac{\lambda_i[g](z)}{\int_{\mathcal{M}} S(\lambda_1[g](z), ..., \lambda_m[g](z)) \, d\nu(z)}) \, d\nu(z) = \mathcal{F}(g).$$

**Proof 2 (Proof of Proposition 2)** *Note that* $\mathrm{Tr}(H_\theta(z)) = \sum_i \lambda_i(z)$ *and* $\mathrm{Tr}(H_\theta^2(z)) = \sum_i \lambda_i^2(z)$, *and denote by* $E := \mathbb{E}_{z \sim P_\phi}[\mathrm{Tr}(H_\theta(z))]$. *Then*

$$\mathbb{E}_{z \sim P_\phi}[\sum_{i=1}^{m} (\frac{\lambda_i(z)}{\mathbb{E}_{z \sim P_\phi}[\sum_i \lambda_i(z)/m]} - 1)^2] = \mathbb{E}_{z \sim P_\phi}[\sum_{i=1}^{m} (\frac{m\lambda_i(z)}{E} - 1)^2]$$

$$= \mathbb{E}_{z \sim P_\phi}[\sum_{i=1}^{m} (\frac{m^2 \lambda_i^2(z)}{E^2} - \frac{2m\lambda_i(z)}{E} + 1)]$$

$$= \mathbb{E}_{z \sim P_\phi}[\sum_{i=1}^{m} \frac{m^2 \lambda_i^2(z)}{E^2}] - m$$

$$= m^2 \frac{\mathbb{E}_{z \sim P_\phi}[\mathrm{Tr}(H_\theta^2(z))]}{\mathbb{E}_{z \sim P_\phi}[\mathrm{Tr}(H_\theta(z))]^2} - m.$$

# B  PSEUDOCODE

In this section, we provide pytorch style pseudocode for our Isometric Regularization and Flattening Module.

## B.1  ISOMETRIC REGULARIZATION OF AUTOENCODERS

In the main manuscript, we mainly study the isometric regularization effect on Variational Autoencoder (VAE). VAE is just one possible choice; our method is straightforwardly applicable to other types of autoencoders. We consider a deterministic encoder and decoder in the following pseudocode.

```python
'''We assume
x: input data w/ size (batch_size, data_space_dimension)
z: latent value w/ size (batch_size, latent_space_dimension)
encoder: torch.nn.Module (e.g., z = encoder(x))
decoder: torch.nn.Module (e.g., x = decoder(z))
eta: mixup parameter
'''
def relaxed_distortion_measure(decoder, z):
    bs, z_dim = z.size()

    v = torch.randn(bs, z_dim)
    Jv = torch.autograd.functional.jvp(decoder, z, v=v)
    TrG = torch.sum(Jv**2, dim=1).mean()
    JTJv = torch.autograd.functional.vjp(decoder, z, v=JV)
    TrG2 = torch.sum(JTJv**2, dim=1).mean()
    return TrG2/(TrG**2)

def isometric_regularization_term(x, encoder, decoder, eta):
    z = encoder(x)
    bs, z_dim = z.size()

    # sample z_augmented from P_Z
    z_permuted = z[torch.randperm(bs)]
    alpha_samples = (torch.rand(bs, 1) * (1 + 2*eta) - eta)
    z_augmented = alpha_samples*z + (1 - alpha_samples)*z_permuted

    # compute relaxed distortion measure
    return relaxed_distortion_measure(decoder, z_augmented)
```

## B.2  FLATTENING MODULE

```python
'''In addition to the above, we assume
z_: new latent value w/ size (batch_size, latent_space_dimension)
flattening_module: torch.nn.Module (e.g., z_ = flattening_module(z)),
    invertible
'''

def flattening_loss(x, encoder, decoder, flattening_module, eta):
    z_ = flattening_module(encoder(x))
    bs, z_dim = z_.size()

    z_permuted = z_[torch.randperm(bs)]
    alpha_samples = (torch.rand(bs, 1) * (1 + 2*eta) - eta)
    z_augmented = alpha_samples*z_ + (1 - alpha_samples)*z_permuted

    def func(z_):
        z = inverse_of_flattening_module(z_)
        return decoder(z)

    return relaxed_distortion_measure(func, z_augmented)
```

## C  COMPARISON TO OTHER REGULARIZATION APPROACHES THAT USE THE JACOBIAN.

There have been several works on regularized autoencoders that use the Jacobian of the encoder or decoder. In the following discussion, let $g(x)$ be an encoder and $f(z)$ be a decoder, and denote the Jacobian of $g(x)$ by $J_g(x)$ and the Jacobian of $f(z)$ by $J_f(z)$. Let $\|\cdot\|_F$ denote the Frobenius norm. The latent space dimension is denoted by $m$. $P_D$ is the data distribution and $P_Z$ is the (augmented) encoded data distribution.

The Contractive Autoencoder (CAE) attempts to enhance robustness of representation by penalizing the Jacobian norm of the encoder function (Rifai et al., 2011b). Mathematically, the regularization term is

$$\mathbb{E}_{x \sim P_D}[\|J_g(x)\|_F^2].$$

The Contractive Autoencoder with Hessian regularization (CAE+H) introduced not long after in (Rifai et al., 2011a) has regularization term

$$\mathbb{E}_{x \sim P_D}[\|J_g\|_F^2 + \gamma \|J_g(x) - J_g(x + \epsilon)\|_F^2],$$

where $\epsilon$ is a small value and $\gamma$ controls the balance between the two terms. The second order regularization using the Hessian penalizes curvature, and thus favors smooth manifolds. These approaches mainly aim to learn robust representations by finding smooth encoders $g$ that have small Jacobian and Hessian norms.

More recently, a regularization method that finds a geometry preserving mapping has been introduced (Chen et al., 2020), called the Flat Manifold Variational Autoencoder (FMVAE). This is the closest work to our paper, and similarly aims to find a scaled isometry. However, the regularization term introduced in FMVAE has several limitations compared to ours.

We first note that geometric objects that we want to preserve, such as length, angle, and volume, are all coordinate invariant concepts, which require a coordinate invariant formulation from the start. If one uses a coordinate-variant measure, then there is no guarantee that it will work effectively for different choices of coordinates. Also, there is no a priori way of knowing which coordinates are best.

In this regard, we first show that the regularization term defined in FMVAE is not coordinate-invariant. Let $\mathcal{M}$ be a Riemannian manifold of dimension $m$ with local coordinates $z \in \mathbb{R}^m$ and Riemannian metric $G(z) \in \mathbb{R}^{m \times m}$, and $\mathcal{N}$ be a Riemannian manifold of dimension $n$ with local coordinates $x \in \mathbb{R}^n$ and Riemannian metric $H(x) \in \mathbb{R}^{n \times n}$. Let $f : \mathcal{M} \to \mathcal{N}$ be a smooth mapping, represented in local coordinates by the italic symbol $f : \mathbb{R}^m \to \mathbb{R}^n$. Let $J_f$ be the Jacobian of $f$ and $P_Z$ be the probability distribution expressed in $\mathbb{R}^m$. Let $\nu$ be a positive measure on $\mathcal{M}$.

The regularization term defined in FMVAE is then

$$\|J_f^T(z)H(f(z))J_f(z) - cG(z)\|_F^2,$$

where $\|\cdot\|_F$ is the Frobenius norm and $c = \int_{\mathcal{M}} \frac{1}{m} \mathrm{Tr}(J_f^T(z)H(f(z))J_f(z))d\nu$. For simplicity, consider a pair of linear coordinate transformations on the input manifold $z' = Az$ and output manifold $x' = Bx$. Then the function $f$ is transformed to $f'(z') := Bf(A^{-1}z')$, $G(z)$ is transformed to $G'(z') := A^{-T}G(z)A^{-1}$ and $H(x)$ is transformed to $H'(x') := B^{-T}H(x)B^{-1}$. Then, after some calculations, the above regularization term is transformed to

$$\left\|A^{-T}\big(J_f^T(z)H(f(z))J_f(z) - c'G(z)\big)A^{-1}\right\|_F^2,$$

where $c' = \int_{\mathcal{M}} \frac{1}{m} \mathrm{Tr}(A^{-T}J_f^T(z)H(f(z))J_f(z)A^{-1})d\nu$. Recall that the Frobenius norm $\|A\|_F^2 = \frac{1}{2}\mathrm{Tr}(A^TA)$, the $A^{-T}$ and $A^{-1}$ multiplied at sides are not canceled; we can see that this is not coordinate-invariant.

In addition to this, perhaps a more direct reason that the FMVAE does not perform as well as our method can be explained as follows. We remark that one of the desired properties of the scaled isometry measure is the third condition in Section 3.3: given two mappings $f$ and $f'$ from $\mathcal{M}$ to $\mathcal{N}$, if $J_f^T(z)H(f(z))J_f(z) = cJ_{f'}^T(z)H(f'(z))J_{f'}(z)$ for some $c > 0$ for all $z$, then the scaled isometry measure should be the same (that is, the metric should not distinguish between $f$ and $f'$).

This makes the measure more natural, in the sense that it does not a priori favor a particular scale for the pullback metric; if the pullback metrics are equivalent up to some scale, then these should be treated the same.

The regularization term introduced in FMVAE can also be viewed as a scaled isometry measure. However, it fails to satisfy the all-important third property. Even though the pullback metrics of $f$ and $f'$ are equivalent up to some scalar multiplication, i.e., $J_f^T(z)H(f(z))J_f(z) = cJ_{f'}^T(z)H(f'(z))J_{f'}(z)$ for some $c > 0$ for all $z$, the one with the "smaller" Jacobian (i.e., smaller norm) achieves a lower value of the regularization term. As a result, the FMVAE favors a mapping with a smaller Jacobian, which can be detrimental to learning accurate data manifolds.

On the other hand, our relaxed distortion measures are defined in a coordinate-invariant way. Further, as we have shown in our main manuscript, the measure does not favor a particular scale of the pullback metric. We believe, because of these characteristics, IRVAE can outperform FMVAE.

# D   EXPERIMENTAL DETAILS

## D.1   SECTION 5.1.1

**Dataset:** We use MNIST dataset. The training, validation, and test data are 50,000, 10,000, and 10,000, respectively.

**Network Architecture:** For both FMVAE, IRVAE, we use fully-connected neural networks that have four hidden layers with ReLU activation functions (256 nodes for each layer are used) for encoder and decoder. The output activation functions are linear and sigmoid for encoder and decoder, respectively. For FM, we use the RealNVP model of depth 8 and length 512.

**Other Details:** For FMVAE and IRVAE, the batch size is 100, the number of training epochs is 300, the learning rate is 0.0001, and $\eta = 0.2$. For FM, the batch size is 100, the number of training epochs is 100, the learning rate is 0.0001, $\beta = 0$, and $\eta = 0.2$. We use Adam optimizer. Validation sets are used to determine optimal models during training.

## D.2   SECTION 5.1.2

**Dataset:** We use CMU motion capture dataset. The training, validation, and test data are 10,000, 2,000, and 2,000, respectively.

**Preprocessing:** In the pre-processing step, the position and orientation of human body center (root) are removed. In addition, the joint angles that have nearly no movements (clavicles, fingers) are removed as done in FMVAE (Chen et al., 2020), and as a result, each pose data is expressed as a 50-dimensional vector.

**Network Architecture:** For VAE, FMVAE, and IRVAE, we use fully-connected neural networks that have two hidden layers with ReLU activation functions (512 nodes for each layer are used) for encoder and decoder. The output activation functions are linear and tanh for encoder and decoder, respectively. For FM, we use the RealNVP model of depth 8 and length 512.

**Other Details:** For VAE, FMVAE and IRVAE, the batch size is 100, the number of training epochs is 300, the learning rate is 0.0001. $\eta = 0.2$ for FMVAE and IRVAE. For FM, the batch size is 100, the number of training epochs is 300, the learning rate is 0.0001, $\beta = 0$, and $\eta = 0.2$. We use Adam optimizer. Validation sets are used to determine optimal models during training.

## D.3   SECTION 5.2

**Dataset:** The number of training, validation, and test data are 162700, 19937, and 19962.

**Network Architecture:** We denote a Con2d layer by Con2d (input channel, output channel, kernel size, stride, padding) and ConvTranspose2d layer by ConvT2d (input channel, output channel, kernel size, stride, padding).

For VAE, FMVAE, and IRVAE, encoders are Convolutional Neural Networks with the following architecture: i) Conv2d (3, 128, 5, 2, 0), ii) Conv2d (128, 256, 5, 2, 0), iii) Conv2d (256, 512, 5, 2,

0), iv) Conv2d (512, 1024, 5, 2, 0), v) Conv2d (1024, 256, 1, 1, 0) with ReLU hidden layer activation functions, and decoders are Convolutional Neural Networks with the following architecture: i) ConvT2d (128, 1024, 8, 1, 0), ii) ConvT2d (1024, 512, 4, 2, 1), iii) ConvT2d (512, 256, 4, 2, 1), iv) ConvT2d (256, 128, 4, 2, 1) , and v) ConvT2d (128, 3, 1, 1, 0) with ReLU hidden layer activation functions. The output activation functions are linaer and tanh for encoder and decoder, respectively.

For FM, we use the RealNVP model of depth 8 and length 512.

For BR, we use the same encoder network as VAE, but add a normalizing layer (i.e., divide the output by its norm) at last (this addition improves the retrieval performance, very significantly).

For ResNet-50, we use the pre-trained model on ImageNet.

**Other Details:** For FMVAE and IRVAE, the batch size is 100, the number of training epochs is 100, the learning rate is 0.0001, and $\eta = 0.2$. For FM, the batch size is 100, the number of training epochs is 100, the learning rate is 0.00001, and $\eta = 0.2$. We use Adam optimizer. For BR, the batch size is 100, the number of training epochs is 100, the learning rate is 0.001 (binary cross entropy loss is used). Validation sets are used to determine optimal models during training.

For FMVAE, we test the following regularization coefficients $\alpha \in \{0.1, 1, 10, 100\}$ and report the best results at $\alpha = 1$. For IRVAE, we test the following regularization coefficients $\alpha \in \{1, 10, 100, 1000\}$ and report the best results at $\alpha = 10$. For FM, we use the stabilizing term with $\beta = \{10, 100\}$ and report the best results at $\beta = 100$; and it is empirically observed that using small learning rate is very important.

Cosine similarity is used as a similarity measure except for the ResNet-50 that uses Euclidean metric (we have tested both cosine similarity and Euclidean metric for each learned representation, and found that cosine similarity mostly works better except the ResNet-50).

# E    ADDITIONAL EXPERIMENTAL RESULTS

## E.1    ADVANTAGES OF ISOMETRIC REGULARIZATION FROM A GENERATIVE PERSPECTIVE

Autoencoders are not only useful for representation learning, but also can be used for realistic data generation. In this section, we show advantages of our isometric regularization from a generative perspective with the following experiments: (i) modulation in the latent space and (ii) linear interpolation in the latent space. We use the CMU motion capture data with the fully connected neural network as described in D.2. We use the latent space of dimension 8.

**Latent Space Modulation :** Figure 6 shows eight generated poses for VAE and IRVAE obtained by the following procedure: (i) encode the original pose in the latent space, (ii) translate the encoded latent value along each latent space axis, and (iii) decode the translated values back to the data space. In case of VAE, translations along the latent space axes except $z_2, z_6$ axes do not generate new poses. In particular, the pose changes really a lot along the $z_2$ axis and yet just a little along the $z_6$ axis. This shows that the trained decoder in VAE is far from the isometry. In contrast, with the isometric regularization in IRVAE, each translated latent value along each latent space axis generates a different pose. Unlike VAE, diverse poses are generated, and in other words, a more disentangled representation is obtained. This property of the isometric decoder has a great advantage (Bengio et al., 2013; Chen et al., 2016).

**Latent Space Interpolation :** Figure 7 shows sequences of poses generated from the latent space linear interpolants between a walking pose and balancing pose. In case of VAE, punching poses suddenly appear in the interpolation of the walking and balancing poses. In contrast, IRVAE does not produce such poses irrelevant to the two given start and end poses. The IRVAE produces more smoothly varying poses than VAE, and the pose interpolation obtained by the IRVAE is closer to the geodesic interpolation along the pose data manifold.

## E.2    DIVERSE IMAGE DATA WITH HIGHER LATENT SPACE DIMENSIONS

We provide additional results with diverse image data (*MNIST, FMNIST, SVHN, CIFAR-10*) using various latent space dimensions. Figure 8 shows that i) the tradeoff curves of IRVAE are located

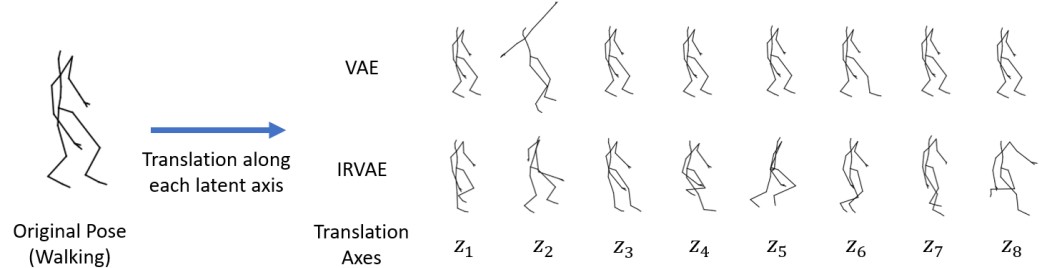

Figure 6: The original pose is encoded in the latent space, then the encoded latent value is translated along each latent space axis ($z_1, z_2, ..., z_8$). The translated latent values are decoded back to generate a new eight pose for each model VAE and IRVAE. The translated distances are proportional to the standard deviations of the encoded training data.

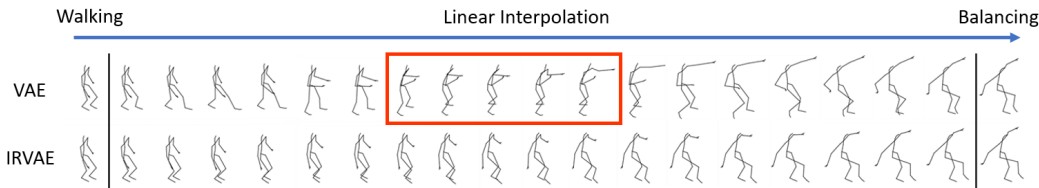

Figure 7: Pose interpolations between a walking pose and balancing pose with linear interpolations in the latent spaces. The red box indicates suddenly appeared punching poses.

lower than those of FMVAE and ii) the FM learns more isometric representations without losses in MSEs.

**Dataset:** The number of training, validation, test data is 50,000, 10,000, 10,000 for MNIST, FM-NIST, SVHN, and 45000, 5000, 10,000 for CIFAR-10.

**Network Architecture:** Let $m$ denotes the latent space dimension. For VAE, FMVAE, IRVAE, encoders are Convolutional Neural Networks with the following architecture: i) Conv2d (3, 32, 4, 2, 0), ii) Conv2d (32, 64, 4, 2, 0), iii) Conv2d (64, 128, 4, 2, 0), iv) Conv2d (128, 256, 2, 2, 0), v) Conv2d (256, 2m, 1, 1, 0) with ReLU hidden layer activation functions, and decoders are Convolutional Neural Networks with the following architecture: i) ConvT2d (m, 256, 8, 1, 0) ii) ConvT2d (256, 128, 4, 2, 1) iii) ConvT2d (128, 64, 4, 2, 1) iv) ConvT2d (64, 3, 1, 1, 0) with ReLU hidden layer activation functions. The output activation functions are linear and sigmoid for encoder and decoder, respectively.

For FM, we use the RealNVP model of depth 8 and length 512.

**Other Details:** For FMVAE and IRVAE, the batch size is 100, the number of training epochs is 100, the learning rate is 0.0001, and $\eta = 0.2$. For FM, the batch size is 100, the number of training epochs is 100, the learning rate is 0.00001, and $\eta = 0.2$. We use Adam optimizer. Validation sets are used to determine optimal models during training.

### E.3 ABLATION STUDY ON MIXUP PARAMETER $\eta$

In the isometric regularization (and flattening module), we use the augmented encoded data distribution $P_Z$ whose sampling procedure is given as follows: (i) encode data $x_i, x_j$ to $z_i, z_j$ by the encoder (and by the invertible map composed with the encoder) and (ii) compute a new latent value $z$ with $z = \delta z_i + (1 - \delta)z_j$ for $\delta$ uniformly sampled form $[-\eta, 1 + \eta]$, where $\eta$ is the mixup parameter.

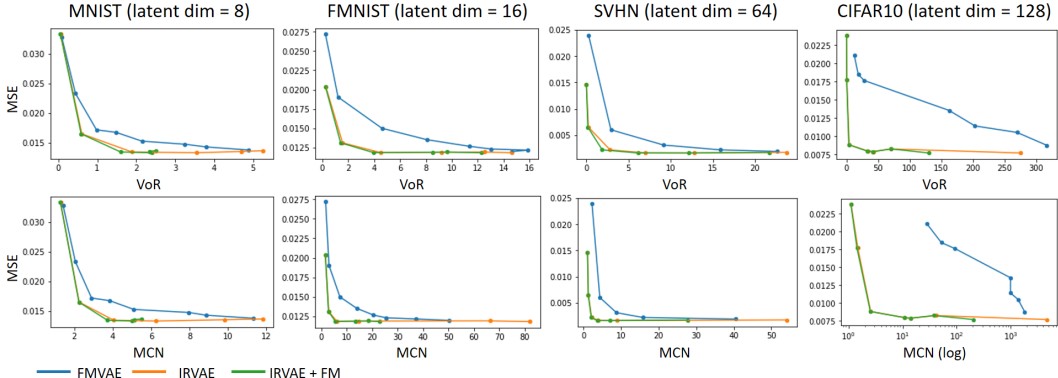

Figure 8: The tradeoff curves of FMVAE, IRVAE, and IRVAE + FM.

Then, our relaxed distortion measure is defined over the support of $P_Z$, i.e.,

$$\mathbb{E}_{z \sim P_Z}\Big[ \sum_i \big(\frac{\lambda_i(z)}{\frac{1}{m}\mathbb{E}_{z \sim P_Z}[\sum_i \lambda_i(z)]} - 1\big)^2\Big],$$

where $f$ is the decoder, $J_f$ is the Jacobian of $f$, and $\lambda_i(z)$ are eigenvalues of $J_f^T J_f(z)$. The mixup augmentation technique is used to extend the influence of isometric regularization to an area where data does not exist.

In this section, we visually analyze the effects of the mixup augmentations in a range of mixup parameters $\eta$. We use the CMU data with fully connected networks described in D.2 and 2-dimensional latent spaces.

Figure 9 shows the latent space data distributions with some equidistance ellipses (the colors of the ellipses represent the condition numbers; redder-the-bigger). In no mixup case, there are many un-isotropic red ellipses in regions between data points, meaning that they are not isometrically regularized. Mixup augmentation makes it possible to regularize even regions between data. When the mixup parameter $\eta = 0$, only the area covered by data interpolation is regularized. By increasing the mixup parameter $\eta$, the outer area, which is covered by data extrapolation, begins to be regularized. Although using bigger eta seems to be unconditionally good because the wider area is regularized, there is indeed a tradeoff as we can see from Figure 9. When the mixup parameter is too big as $\eta = 1$, some inner area (i.e., area covered by data interpolation) is not sufficiently regularized because of the cost of regularizing the outer domain. We use a balanced value $\eta = 0.2$ for all experiments conducted in this paper (for all algorithms IRVAE, FM and FMVAE).

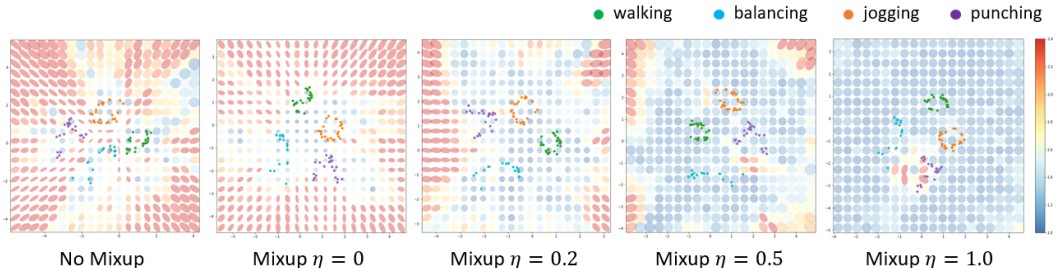

Figure 9: The effect of mixup augmentation with varying mixup parameters $\eta$.

### E.4 COMPUTATIONAL SPEED

In this section, we provide per-epoch runtimes of VAE, FMVAE, IRVAE, and FM. We use the MNIST data and same experiment settings in D.1. We use the GeForce RTX 3090 for GPU re-

sources. Table 2 shows the per-epoch runtimes. The difference in computation times arises from the computations of the regularization terms. The IRVAE takes a little longer than the FMVAE because, while the regularization term in FMVAE requires to compute the Jacobian vector product only, the regularization term in IRVAE requires to compute both the Jacobian vector product and vector Jacobian product. In the case of FM, it takes much longer because the invertible neural network architecture is relatively heavier. Fortunately, in training of FM that uses the pre-trained IRVAE, we empirically observed that only a few epochs are required for convergence, probably because the IRVAE already had learned isometric representations to a certain degree.

Table 2: Averages and standard deviations of the per-epoch runtimes. 50,000 MNIST image training data are used with 100 batch size. For VAE, FMVAE, IRVAE, we use the 4 layers of fully connected neural networks, and for FM, we use the RealNVP of depth 8.

|  | VAE | FMVAE | IRVAE | FM |
|---|---|---|---|---|
| per-epoch runtime | $1.91 \pm 0.0818$ s | $2.80 \pm 0.0755$ s | $2.81 \pm 0.0869$ s | $19.9 \pm 0.235$ s |

### E.5 ISOMETRIC REGULARIZATION FOR OTHER AUTOENCODERS

Our isometric regularization algorithm can be used with other types of autoencoders although we focus on the Variational Autoencoder (VAE) in the main manuscript. In this section, we verify the effects of the isometric regularization with more diverse autoencoder methods: Vanilla Autoencoder (AE) (Kramer, 1991), Wasserstein Autoencoder (WAE) (Tolstikhin et al., 2018), and Denoising Autoencoder (DAE) (Vincent et al., 2010).

We use the MNIST data and same experiment settings in D.1. For DAE, we use the Gaussian noise with standard deviation of 0.01. For WAE, we use the maximum mean discrepancy, median heuristic for bandwidth selection, and the regularization coefficient 0.001.

Figure 10 shows the effects of the isometric regularization to diverse autoncoders (AE, VAE, DAE, WAE). Regardless of the autoencoder types, we consistently observe that the added isometric regularization terms effectively separate data of different classes (even without major tuning of the regularization coefficient $\alpha$).

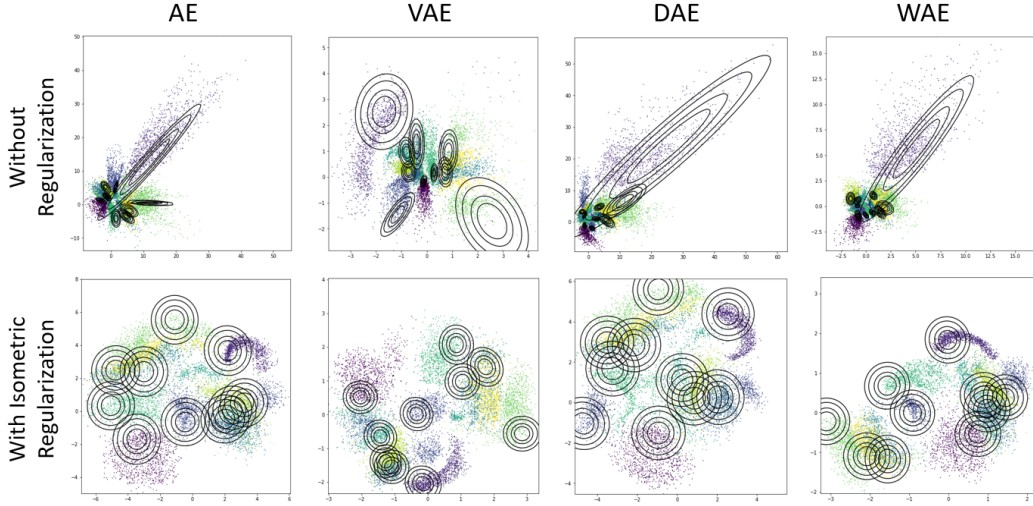

Figure 10: Isometric regularization with diverse autoencoder methods. The more homogeneous and isotropic equidistance plots are, the more isometric the representations are.

### E.6  Isometry vs Scaled Isometry

In this section, we show empirical results that support our claim in the introduction that finding a scaled isometry is better than finding a strict isometry.

Whereas an isometry exactly preserves angles and distances, a scaled isometry preserves angles and scaled distances. At first this may seem a superficial difference – why not simply choose length scales so that the two spaces have the same scales? – but the reason this difference is consequential for our problem is that we do not a priori have a precise charcterization (and hence their relative scales) of the two spaces. We are in effect discovering the manifold structure of the data space while constructing a latent space representation for the data manifold at the same time.

To characterize the above more precisely, let $f(z)$ be a decoder, $J_f(z)$ be the Jacobian of $f(z)$, and $P_Z(z)$ be the augmented encoded data distribution. Let $\lambda_i(z)$ be eigenvalues of $J_f^T(z)J_f(z)$. Let $G(z)$ be the latent space metric, and $H(x) = I$. Recall that the isometry $f$ needs to satisfy $J_f^T(z)J_f(z) = G(z)$ for all $z$ in the support of $P_Z$, whereas the scaled isometry $f$ needs to satisfy $J_f^T(z)J_f(z) = cG(z)$ for some $c > 0$ and all $z$ in the support of $P_Z$. To find a strict isometry, we need to set a scalar parameter $k$ that determines the latent space metric, i.e., $G(z) = kI$, in advance, whereas, to find a scaled isometry, we just need to set $G(z) = I$ as done in the paper.

To find the isometry, we add the following distortion measure as a regularization term:

$$\alpha \mathbb{E}_{z \sim P_Z}\Big[\sum_i (\lambda_i(z) - k)^2\Big],$$

where $k$ is the pre-defined parameter and $\alpha$ is the regularization coefficient. We need to search for the optimal parameter $k$ in this case.

In contrast, we do not need such parameter $k$ to find a scaled isometry. We just need to add the following relaxed distortion measure as a regularization term:

$$\alpha \mathbb{E}_{z \sim P_Z}\Big[\sum_i \big(\frac{\lambda_i(z)}{\frac{1}{m}\mathbb{E}_{z \sim P_Z}[\sum_i \lambda_i(z)]} - 1\big)^2\Big],$$

where $\alpha$ is the regularization coefficient.

Figure 11 shows the tradeoff curves (lower-the-better) obtained by using (i) the relaxed distortion measure (for finding a scaled isometry) or (ii) the distortion measure with a range of user-defined parameters $k$ (for finding a strict isometry). For the strict isometry learning case, it can be seen that the performance peaks when the value of $k$ is at an appropriate level that is neither too high nor too low. In this experiment, the optimal $k$ lies in between 10 and 1000 and can be experimentally found only by performing a finer grid search. The tradeoff curves obtained by using the relaxed distortion measure are always located lower than the tradeoff curves obtained by using the distortion measures. From this experimental result, we conclude that it is better to find a scaled isometry than strict isometry.

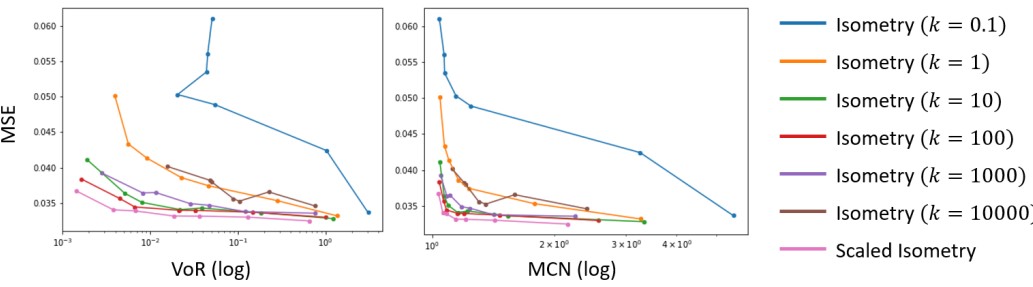

Figure 11: Tradeoff curves obtained by changing the regularization coefficients $\alpha$ (lower-the-better).

E.7 MULTIPLE TIMES RUN TO COMPUTE AVERAGES AND STANDARD DEVIATIONS

It is important to judge whether the experimental result is a result of randomness or not. In this section, we take the same experiments done in Figure 1 and 3 of the main manuscript with MNIST and CMU data multiple times, and report the averaged tradeoff curves and standard deviations of the measured metrics.

Figure 12 shows the results, where the standard deviations computed over multiple experiments with different random seeds are visualized as ellipses. The horizontal axes lengths of the ellipses represent the standard deviations of the VoR and MCN, and the vertical axes lengths of the ellipses represent the standard deviations of the MSE. It is correct to use the standard errors to see the standard deviations of the sample means, i.e., standard deviations divided by the square root of the number of experiments, but we use the standard deviations since the standard errors are too small to visualize. The sizes of the ellipses are rally small, which provides a solid evidence that IRVAE outperforms FMVAE.

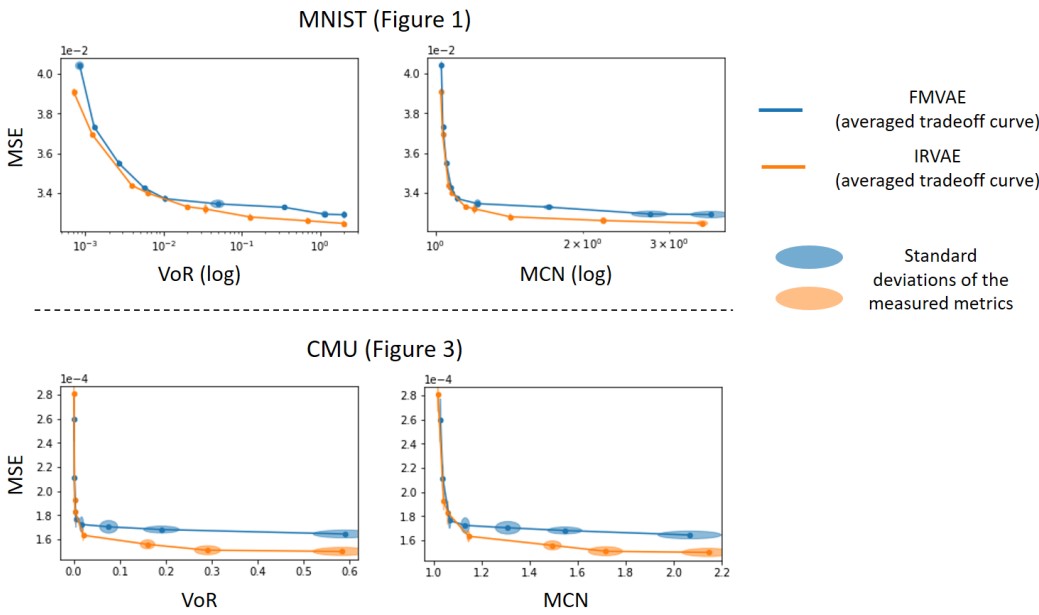

Figure 12: Averaged tradeoff curves and standard deviations represented as ellipses for MNIST and CMU experiments in Figure 1 and 3 of the main manuscript (20 times run). We wanted to draw ellipses by using the standard errors, but they were too small to visualize. Even standard deviations are really small.

E.8 SENSITIVITY ANALYSIS OF THE LATENT SPACE DIMENSION

In general, when training an autoencoder, the most important parameter is the latent space dimension. Although there have been some works on estimating the intrinsic dimension of the data distribution (Levina & Bickel, 2005), given a complex high-dimensional data, the intrinsic dimension estimation problem is very challenging; in practice, an appropriate value is often found, empirically. Throughout literature, there are some frequently-used choices for well-known standard image datasets. For relatively simple datasets such the MNIST and FMNIST, latent space dimensions $2 \sim 32$ are often used, and for more complex and higher dimensional datasets such as the CIFAR10 and CELEBA, latent dimensions $64 \sim 512$ are often used.

In this section, using the MNIST image data with the fully connected network described in D.1 (trained for 100 epochs), we provide an analysis of the isometric regularization in autoencoders with multiple latent space dimensions. Table 3 shows averages and standard deviations of the condition numbers of the pullback metrics $J_f^T(z)J_f(z)$ ($f$ is the decoder and $J_f$ is the Jacobian of $f$) computed

with the test data. Both the averages and standard deviations should be small if $J_f^T(z)J_f(z) = cI$ for some $c > 0$ for all encoded test data $z$.

In the case of VAE, it can be seen that both the average and standard deviation of the condition number sharply increase as the latent dimension increases. On the other hand, IRVAE appears to be able to effectively find an isometric representation even if the dimension of the latent space increases. In the case of 16-dimensional latent space, the condition number of IRVAE is about 32 which is not small enough, and this result may imply that the true intrinsic dimension of the MNIST data manifold is lower than 16.

In summary, our isometric regularization can effectively find more isometric representations than vanilla autoencoders regardless of the latent space dimensions. When the dimension of the latent space is too large, it may not be possible to obtain a sufficiently isometric representation space. This This seems because the dimension of the ground true manifold is much smaller than the selected latent space dimension.

Table 3: Averages and standard deviations of the condition numbers of the pullback metrics for VAE and IRVAE with respect to the latent space dimension. The lower the average and standard deviations are, the more isometric the latent is.

| Latent Space Dimension | VAE | IRVAE |
|:---:|:---:|:---:|
| 2 | $5.39 \pm 4.72$ | $1.41 \pm 0.55$ |
| 8 | $85.55 \pm 85.69$ | $1.87 \pm 0.25$ |
| 16 | $1635018.88 \pm 1789834.25$ | $37.28 \pm 42.12$ |

# F   DETAILED RESULTS FOR SINGLE ATTRIBUTE RETRIEVAL

Table 4 shows the detailed retrieval results (P@10 of the single attribute retrieval). Our algorithms (IRVAE and IRVAE + FM) have much higher performance than FMVAE and VAE, and even show performance close to the supervised method (BR).

Table 4: Single attribute retrieval results of Precision at 10, P@10 (for total 40 attributes). The best results among unsupervised methods are colored red, and the best results among all six methods are marked bold.

| Attributes | VAE | FMVAE | IRVAE (ours) | IRVAE + FM (ours) | Pre-trained (ResNet-50) | Supervised (BR) |
|---|---|---|---|---|---|---|
| Young | 0.8 | **1** | 0.7 | 0.8 | 0.7 | **1** |
| Wearing Necktie | 0.4 | **0.7** | 0.6 | **0.7** | 0.1 | 0.5 |
| Wearing Necklace | 0.2 | 0.1 | **0.3** | 0.2 | 0.1 | 0.1 |
| Wearing Lipstick | **1** | 0.9 | **1** | **1** | 0.4 | **1** |
| Wearing Hat | 0.4 | 0.7 | 0.8 | 0.9 | 0 | **1** |
| Wearing Earrings | 0.5 | 0.5 | 0.7 | 0.6 | 0.4 | **1** |
| Wavy Hair | 0.8 | **0.9** | **0.9** | **0.9** | 0.3 | **0.9** |
| Straight Hair | **0.9** | 0.2 | 0.7 | 0.7 | 0.2 | 0.5 |
| Smiling | **1** | **1** | **1** | **1** | 0.7 | **1** |
| Sideburns | 0.4 | 0.7 | 1 | 0.9 | 0.2 | 0.6 |
| Rosy Cheeks | 0.6 | 0.5 | 0.8 | 0.7 | 0 | **1** |
| Receding Hairline | 0.3 | **0.8** | 0.6 | 0.5 | 0 | **0.8** |
| Pointy Nose | 0.5 | 0.6 | 0.4 | **0.8** | 0.2 | **0.8** |
| Pale Skin | 0.3 | **0.8** | 0.7 | 0.7 | 0.1 | **0.8** |
| Oval Face | 0.5 | **0.6** | 0.5 | 0.4 | 0.5 | **0.6** |
| No Beard | 0.8 | **1** | **1** | **1** | 0.7 | **1** |
| Narrow Eyes | 0.5 | 0.1 | 0.3 | 0.5 | 0.1 | **0.7** |
| Mustache | 0.2 | 0.3 | 0.3 | **0.4** | 0.1 | 0.3 |
| Mouth Slightly Open | **1** | **1** | **1** | **1** | 0.6 | **1** |
| Male | **1** | **1** | **1** | **1** | 0.4 | **1** |
| High Cheekbones | **1** | **1** | 0.9 | **1** | 0.4 | **1** |
| Heavy Makeup | **1** | 0.8 | **1** | **1** | 0.3 | **1** |
| Gray Hair | 0.5 | 0.7 | 0.7 | **0.8** | 0 | 0.7 |
| Goatee | 0.5 | 0.6 | 0.7 | **0.9** | 0 | 0.7 |
| Eyeglasses | **1** | 0.9 | **1** | **1** | 0.1 | **1** |
| Double Chin | 0.4 | 0.3 | 0.6 | 0.8 | 0.1 | **1** |
| Chubby | 0.3 | 0.1 | 0.3 | 0.4 | 0 | **0.9** |
| Bushy Eyebrows | 0.6 | 0.5 | **1** | **1** | 0.3 | 0.9 |
| Brown Hair | 0.5 | 0.3 | **0.8** | 0.6 | 0.2 | **0.8** |
| Blurry | 0 | 0.1 | 0.2 | 0.1 | 0.1 | **0.4** |
| Blond Hair | **1** | **1** | **1** | **1** | 0.1 | 0.9 |
| Black Hair | 0.4 | 0.9 | 0.9 | 1 | 0.2 | 0.8 |
| Big Nose | 0.4 | **0.6** | 0.5 | 0.5 | 0.1 | **0.6** |
| Big Lips | 0.4 | **0.6** | 0.2 | 0.5 | 0.3 | 0.3 |
| Bangs | **1** | 0.8 | **1** | **1** | 0 | **1** |
| Bald | 0.3 | 0.4 | 0.6 | 0.8 | 0 | **0.9** |
| Bags Under Eyes | 0.5 | 0.4 | 0.6 | 0.7 | 0.4 | **0.7** |
| Attractive | **1** | **1** | 0.9 | 0.9 | 0.5 | **1** |
| Arched Eyebrows | 0.8 | 0.7 | 0.7 | 0.8 | 0.2 | **0.9** |
| 5 o'clock Shadow | **0.7** | 0.5 | **0.7** | 0.6 | 0.3 | 1 |
| Number of REDS | 14 | 15 | 18 | 25 | - | - |
| Number of BOLDS | 12 | 15 | 15 | 19 | 0 | 28 |
| Average Precision | 61.0 | 64.0 | 71.5 | 75.3 | 23.5 | 80.3 |

