# OpenReview forum: "Regularized Autoencoders for Isometric Representation Learning"
_ICLR.cc/2022/Conference — ICLR 2022 Poster_

### Official Review · Reviewer_i9fR · 2021-11-01

**Correctness:** 3
**Technical Novelty And Significance:** 3
**Empirical Novelty And Significance:** 2
**Recommendation:** 6
**Confidence:** 4

**Main Review:**

- The paper shares an insight that whenever the mapping between latent space and data space in a generative model (e.g. VAE) approximates an isometry (or a scaled isometry, as proposed), one observes better results in a range of experiments. I am missing the key motivation behind this, perhaps also due to the paper itself mentioning (beginning of page 2) that this kind of requirement is too much to ensure good reconstruction. What is the rationale behind this? One unclear aspect to me springs from the fact that, in general, an isometry does not even exist between arbitrary manifolds, for example, one can not isometrically embed a sphere into a Euclidean space of any dimension. Why should a quasi-isometry lead to better generative models? Why isn't a homeomorphism enough?

- What I think is really missing, is a so called "killer application" that would convincingly show the benefits of the proposed regularization. In particular, it is important to demonstrate that isometric representation learning is relevant, and that it can lead to significant improvements in many contexts. By contrast, the current experiments show some good results on simple data. Without a strong motivation, or a strong argument in support of the proposed method, it is hard to judge how potentially impactful it can be within the broader research field.

- I would expect the choice of dimensions D and m to be important, since the distortion measure can vary greatly among different embeddings. How were these parameters chosen? I suggest to include some sensitivity experiments showing the influence of these dimensions on the final results, together with a discussion on their effect. Figure 6 in the supplementary goes in this direction, but I am missing an interpretation of these plots. Similarly, I could not find a sensitivity experiment showing the influence of parameter k (degree of the conformal mapping).

- What is the consequence of choosing the identity metric for the data space?

- The proposed measure (eq.7) is novel to my knowledge and is well described.

- The manuscript is very pleasant to read, it is fluent, solid, and clear from start to end. The mathematics are sound and accessible. The writing is sharp.

- The idea behind the paper is novel to my knowledge, but I found a few related works in recent conferences, see e.g. "LIMP: Learning Latent Shape Representations with Metric Preservation Priors" (ECCV 2020), it might be worth checking these out.

A few side-comments:

- It would be nice, although not strictly necessary, to visually illustrate the difference between scaled isometries of different order, perhaps by texture mapping a checkerboard pattern between 2D surfaces embedded in 3D if at all possible.

- How costly is the flattening step (solving eq.11)?

- How should the equidistance plot be read? The more isotropic, the better?

- The paper mentions a few time some "popular" choices for the h() function. To what literature does the popularity refer to?

- There is an incomplete sentence in page 5 "inherent tradeoff We introduce".

- At the end of page 5, "the the".

**Summary Of The Paper:**

The paper makes two main contributions. The first is the introduction of a new regularizer term for the VAE loss, ensuring a (scaled) isometry between the learned latent space and the (typically unknown) data space. The second is a post-processing "flattening" step to improve the isometry constraint by directly operating on the latent space, while leaving the reconstruction error untouched. The resulting pipeline seems effective, as showcased on a selection of experiments over standard datasets.

**Summary Of The Review:**

A well-written manuscript with a practical implementation of an isometric representation learning pipeline, which is nonetheless counterbalanced by unclear motivations and not very impressive results. Might still have an impact on a certain community, but a potentially broader impact is not clear at this stage.

---

> ### Author Response · Authors · 2021-11-17
> **Response to Reviewer i9fR (3)**
>
> __Q5.__ The idea behind the paper is novel to my knowledge,
> but I found a few related works in recent conferences, see e.g.
> "LIMP: Learning Latent Shape Representations with Metric
> Preservation Priors" (ECCV 2020), it might be worth checking these out.
>
> __A5.__ Thank you for the reference. We will certainly
> examine this reference and others cited in it, and if appropriate,
> cite these in the revised version of our manuscript.
>
> __Q6.__ How costly is the flattening step (solving eq.11)?
>
> __A6.__ ``We have added additional experiments
> in Appendix E.4 that compare the computational requirements of our
> algorithm against the baselines``. In our experience, only a
> few epochs are needed for convergence.
>
> __Q7.__ How should the equidistance plot be read? The more
> isotropic, the better?
>
> __A7.__ Yes, the more homogeneous and more isotropic, the
> better. We have added this remark in the revised manuscript.
>
> __Q8.__ The paper mentions a few times some ''popular" choices
> for the h() function. To what literature does the popularity refer
> to?
>
> __A8.__ The main reference we have in mind is the paper
> "A Riemannian geometric framework for manifold learning of
> non-Euclidean data" by Jang, Noh, and Park (cited in our
> manuscript) as well as additional references cited in this paper.
>
> __Q9.__ There is an incomplete sentence in page 5 "inherent
> tradeoff We introduce". At the end of page 5, "the the".
>
> __A9.__ Thank you for catching this error, we have revised
> the sentence accordingly.

---

> ### Author Response · Authors · 2021-11-17
> **Response to Reviewer i9fR (2)**
>
> __Q2.__ What I think is really missing, is a so called
> "killer application" that would convincingly show the benefits
> of the proposed regularization. In particular, it is important
> to demonstrate that isometric representation learning is relevant,
> and that it can lead to significant improvements in many contexts.
> By contrast, the current experiments show some good results on
> simple data. Without a strong motivation, or a strong argument
> in support of the proposed method, it is hard to judge how
> potentially impactful it can be within the broader research field.
>
> __A2.__
> We appreciate the comment, and although we did try to make
> the case both qualitatively and mathematically that isometric
> representation learning is important at several levels, as you
> point out, a strong application certainly helps. We believe
> we have described such applications; perhaps the reported benefits
> may not be up to the reviewer's expectations, but we do believe
> that the gains are meaningful and significant. We've tried to
> demonstrate the advantages of our algorithm for the following
> two practical scenarios:
> * Given a set of unlabelled high-dimensional data, compute
> the similarity between data (distance between data);
> * Using a generative model for data interpolation and modulation.
>
> __Computing similarity between data:__ When given a set
> of unlabelled high-dimensional data, quite possibly the most
> effective way to compute the distance between data is (i) to learn
> the data manifold, and (ii) to compute the geodesic distance
> between data along the learned manifold.  As is well known,
> geodesic computations are very costly and impractical for most
> applications except for the simplest manifolds.  A more
> practical alternative is to seek an isometric representation space
> endowed with a Euclidean metric, so that Euclidean distances computed
> in the latent space closely match geodesic distances in the
> original data manifold. As demonstrated by the results of our experiments
> on human face retrieval using CELEBA data, our algorithms
> outperform the baselines by a significant margin. The performance
> results of our method are in fact quite close to those attained
> by supervised methods.
>
> __Data interpolation and modulation:__
> As is well known, trained decoders are widely used as data generators,
> and one of the characteristics of a good data generator, particularly
> for data interpolation and modulation, is that a point in the input
> latent space that is displaced by a certain amount should produce
> a proportional displacement in the output space. Unfortunately
> vanilla AEs and the VAE often fail to satisfy this property, e.g.,
> small displacements along some latent space axes may produce large
> displacements in the output space, while the opposite may occur for
> other directions. Our isometric decoder is designed to have proportional
> displacements in the input and out spaces, and is well-suited for
> applications that require interpolation and modulation.
> ``We have added additional experimental
> results in Appendix E.1 showing these advantages``.
>
> __Q3.__ I suggest to include some sensitivity experiments
> showing the influence of the latent space dimensions on the final
> results, together with a discussion on their effect.
>
> __A3.__ Thank you for the suggestion. ``We have
> added additional experiments in Appendix E.8 describing results
> of a sensitivity analysis on the latent space dimension.``
>
> __Q4.__ What is the consequence of choosing the identity
> metric for the data space?
>
> __A4.__
> The choice of metric $H(x)$ on the data space (for the autoencoder,
> this metric is used for both the input to the encoder and the output
> of the decoder) is entirely arbitrary; all the definitions and
> constructions developed in the paper still apply. In fact this is
> one of the important benefits of using our coordinate-invariant
> geometric formulation, as the notion of isometry, scaled isometry,
> etc., all remain valid (and invariant) with respect to different
> choices of local coordinates and metrics. If for some particular
> application or dataset it makes more sense to use some other
> metric $H(x)$ rather than the identity, then our framework can be
> applied with no modification.

---

> ### Author Response · Authors · 2021-11-17
> **Response to Reviewer i9fR (1)**
>
> Thank you very much for your constructive feedback, and our sincerest
> apologies for not uploading our comments sooner. In response to the
> many constructive suggestions we have received, we have spent the
> past week conducting a wide range of new experiments, and revising
> our manuscript accordingly. In an attempt to answer the reviewer
> questions, and to better clarify and validate our contributions,
> the appendix has been substantially expanded with the following
> additional content:
>
> * Appendix B: Pseudocode;
> * Appendix C: Comparison to other regularization approaches that use
> 	the Jacobian;
> * Appendix E.1: Advantages of isometric regularization from a
> 	generative perspective;
> * Appendix E.3: Ablation study on mixup parameter;
> * Appendix E.4: Computational speed;
> * Appendix E.5: Isometric regularization for other autoencoders;
> * Appendix E.6: Isometry vs scaled isometry;
> * Appendix E.7: Multiple runs for computing standard deviations;
> * Appendix E.8: Sensitivity analysis of the latent space dimension.
>
> Below we provide detailed responses to each of the reviewer comments.
> When referencing any major changes and addition of new content to the
> revised manuscript, we have indicated those passages ``like this``.
>
> __Q1.__ The paper shares an insight that whenever the mapping
> between latent space and data space in a generative model (e.g. VAE)
> approximates an isometry (or a scaled isometry, as proposed), one
> observes better results in a range of experiments. I am missing the
> key motivation behind this, perhaps also due to the paper itself
> mentioning (beginning of page 2) that this kind of requirement is
> too much to ensure good reconstruction. What is the rationale
> behind this?
>
> __A1.__ We can certainly sympathize with the fact that
> the terms "isometry" and "scaled isometry" are similar and
> hard to distinguish.  Whereas an isometry exactly preserves
> angles and distances, a scaled isometry preserves angles and
> scaled distances. At first this may seem a superficial difference
> -- why not simply choose length scales so that the two spaces
> have the same scales? -- but the reason this difference is
> consequential for our problem is that we do not a priori
> have a precise charcterization (and hence their relative scales)
> of the two spaces. We are in effect discovering the manifold
> structure of the data space while constructing a latent space
> representation for the data manifold at the same time.
>
> To characterize the above more precisely in terms of the Riemannian
> metric on the latent space, one must know in advance the value
> of the scale parameter $k>0$ of the latent space metric,
> and set the latent space metric to $G(z)=kI$ in order to
> construct a strict isometry. Since this is not possible, our
> contention is that the next best thing is to search for a
> scaled isometry, which eliminates the scale factor $k$.
> ``We have further
> clarified these differences between seeking a strict isometry versus a
> scaled isometry in the newly added Appendix E.6.``

---

### Official Review · Reviewer_MiWx · 2021-11-01

**Correctness:** 4
**Technical Novelty And Significance:** 3
**Empirical Novelty And Significance:** 2
**Recommendation:** 5
**Confidence:** 3

**Main Review:**

The authors address an important problem in unsupervised learning, namely, manifold learning using a NN. The method proposed in the paper extends two recent works by Chen et al. and Jang et al. They propose an Isometric Regularization that is realized based on Hutchinson’s stochastic trace estimator to enable efficient estimation of the gradients. The English level is good, but the description of the method is hard to follow. It would be helpful if the authors could add a pseudocode of the proposed methodology. The authors demonstrate that the proposed method leads to several benefits compared to the existing method. However, I have several concerns about the proposed method. Specifically, my major comments are:
-The method proposed here is a variational autoencoder; why is it presented in the introduction as an autoencoder?
-On the same point, the variational part is not well justified; I had to read Chen et al. to understand why the variational part is included.
-I am not convinced that the method leads to any benefit on MNIST. The MSE gain compared to FMVAE is really marginal and could result from non-optimal hyperparameter tuning.
-How are the hyperparameters of all methods tuned?
-If I understand correctly, the latent dimension used in the paper is 2; can you use a higher dimension? I would expect that for MNIST and other image datasets, a higher latent dimension is required.
-The equidistance ellipses are not well defined, are these computed in the ambient space and plotted in the latent space? If this is the case, I am not sure that preserving large Euclidean distances is “good”. In high dimensional space, we should mostly trust small distances and not large ones. Therefore, these ellipses should be related to a small distance in the ambient space. The scale of these ellipses is not provided.
-The improvements in the retrieval task do seem substantial; this supports the effectiveness of the method. However, the way the hyperparameters of all methods are tuned is not explained.
Minor comments:
-”Learning the data manifold”- this term is not well defined in the paper. Perhaps the authors mean “preserve the data manifold”. This is not clear; learning the manifold suggests learning its underlying parametric representation.
 -Before the words “We introduce” on page 4, a period or comma is missing; also, the sentence does not flow.
-Figures 1 and 2, the MSE vs. MCN and MSE vs. VoR from both of these plots are redundant; you could only leave the ones in figure 2; the information is redundant.
-What are the colors in the embedding? The labels? If so, the separation is not improved between classes.
-Please mention in the caption of figure 2 on what data is this experiment evaluated.
-The condition number calculation for figure 4 is not clearly explained; please add some intuition.
The following works have very similar ideas and should be cited:


[1] McQueen, J., et al.  (2016). Nearly isometric embedding by relaxation. Advances in Neural Information Processing Systems, 29, 2631-2639.‏
[2] Peterfreund, Erez, et al. Local conformal autoencoder for standardized data coordinates. Proceedings of the National Academy of Sciences, 2020, 117.49: 30918-30927.‏

**Summary Of The Paper:**

The authors propose a new type of regularized autoencoder (actually variational autoencoder) that is designed to preserve the geometry of the data. They demonstrate that preserving angles and relative distances lead to an improved representation of the data. Specifically, they add a regularization and explore the tradeoff between reconstruction and geometry preservation. They further propose a scheme to flatten the latent representation in a postprocessing fashion.

**Summary Of The Review:**

To summarize, the authors address an important problem and propose a new method for learning a low-dimensional representation from high-dimensional data. They rely on the manifold assumption and propose a new regularized to a variational autoencoder. The results demonstrate that the method leads to improvement in retrieval tasks. However, the method has some limitations (namely the low dimensional embedding dimension), and the experiments are not backed up by a rigorous scheme for hyperparameter tuning. For these reasons and the marginal improvements demonstrated in figures 1-3, I recommend a weak rejection of the paper.

---

> ### Author Response · Authors · 2021-11-17
> **Response to Reviewer MiWx (2)**
>
> __Q3.__ The equidistance ellipses are not well-defined, are
> these computed in the ambient space and plotted in the latent
> space? If this is the case, I am not sure that preserving large
> Euclidean distances is “good”. In high-dimensional space, we
> should mostly trust small distances and not large ones. Therefore,
> these ellipses should be related to a small distance in the
> ambient space. The scale of these ellipses is not provided.
>
> __A3.__ Thank you for pointing this out. In the
> revision ``we have added a more detailed description of the
> equidistance plots``: for a selected point $z_c$, the equidistance
> plot is {$z|(z-z_c)^T J_f^T(z_c) J_f(z_c) (z-z_c) = k$} for $k>0$.
> The more homogeneous and isotropic the ellipoids are, the closer
> the decoder $f$ is to being a scaled isometry (the relative sizes
> of the ellipsoids are more important, the absolute sizes less so).
> By comparing subfigures A, B, F, I, in Figure 1, it is visually
> apparent that subfigures F and I have more homogeneous and
> isotropic ellipsoids compared to A and B, which is also
> quantitatively verified in the upper-left graph.
>
> __Q4.__ If I understand correctly, the latent dimension
> used in the paper is 2; can you use a higher dimension? I would
> expect that for MNIST and other image datasets, a higher
> latent dimension is required.
>
> __A4.__ We deliberately used two-dimensional latent spaces to
> visualize the equidistance plots. Similar experiments with more diverse
> image data (MNIST, FMNIST, SVHN, CIFAR10) and higher-dimensional latent
> spaces have been added in Appendix E.2.  For retrieval tasks, the
> latent space dimension is 128.
>
> __Q5.__ The method proposed here is a variational autoencoder;
> why is it presented in the introduction as an autoencoder?
>
> __A5.__ VAE is just one possible choice; our method is
> straightforwardly applicable to other types of autoencoders.
> ``We have added additional experiments in
> Appendix E.5 that show the effects of our proposed regularization
> term when applied to other autoencoders``.
>
> __Q6.__ How are the hyperparameters of all methods tuned?
> The improvements in the retrieval task do seem substantial; this
> supports the effectiveness of the method. However, the way the
> hyperparameters of all methods are tuned is not explained.
>
> __A6.__ For Figures 1-3, since we compare only the tradeoff
> curves, there is no hyperparameter to be tuned (the mixup
> parameter is fixed to $0.2$ in all cases for fair comparison).
> In the retrieval experiments described in Appendix D, we search
> over the regularization coefficients $\alpha \in$
> {0.1, 1, 10, 100} for FMVAE, while for IRVAE, we search over
> the regularization coefficients $\alpha \in$ {1, 10,
> 100, 1000}.  For FM, we search over $\beta \in$ {10, 100}.
>
> __Q7.__ ”Learning the data manifold”- this term is not well
> defined in the paper. Perhaps the authors mean “preserve the data
> manifold”. This is not clear; learning the manifold suggests
> learning its underlying parametric representation.
>
> __A7.__
> The expressions "learning the data manifold'' and
> "learning latent coordinates that preserve the geometry of the
> learned manifold" mean different things. By the former we mean
> learning its underlying parametric representation --- imagine,
> for example, fitting a multidimensional surface to a set of
> points in some high-dimensional space --  while by the latter
> we mean constructing a coordinate chart for the learned manifold
> that is as distortion-free as possible. We will try to clarify
> the usage of these expressions in the revised manuscript.
>
> __Q8.__ The condition number calculation for Figure 4 is
> not clearly explained; please add some intuition.
>
> __A8.__ The condition number is the ratio of the maximal
> and minimal eigenvalues of $J_f(z)^T J_f(z)$. Intuitively,
> this is the ratio between the square root of the lengths of the
> major and minor axes of the ellipse, with a condition number of
> 1 indicating a perfect circle.
>
> __Q9.__ The following works have very similar ideas and
> should be cited: [1], [2].
>
> __A9.__ We appreciate the references; these will be included
> in the revised manuscript.

---

> > ### Comment · Reviewer_MiWx · 2021-11-23
> > **Response to authors**
> >
> > I have read the response and changes made in the paper, and appreciate the effort done by the authors, the paper has been improved. I still have three minor concerns:
> > 1) There is still a redundancy between figures 1 and 2. I believe I was misunderstood. In both figures, the authors present tradeoffs between MSE and VoR/MCN. The only difference is the added evaluation of FM+IRVAE. This could be presented in one set of two subfigures, instead of four.
> > 2) Citations were not added.
> > 3) Following on A.3-> I see the effect on the ellipsoid. But do not see a consistent effect on the classes structure. In this example, it is hard to interpret if this effect is beneficial for learning the "correct" manifold. Classes are mixed, and other classes are split. Perhaps it is hard to embed MNIST into 2 dimensions using this method. I would suggest using a simpler example to demonstrate the effect of the method, for example: a subset of classes in MNIST, or using a few digits and augmenting them to create an artificial manifold with a known structure.

---

> > > ### Author Response · Authors · 2021-11-23
> > > **Response to Reviewer MiWx**
> > >
> > > A1.  Sorry, we misunderstood your earlier question. We agree that Figure 2 can be merged into Figure 1. Thanks for the nice comment, we will revise this in the final version.
> > >
> > > A2. Citations are now added!
> > >
> > > A3. While this may already be apparent to the reviewer, we'd first like to point out that the geometry regularization term is not beneficial for learning the "correct" manifold. Rather, as demonstrated in the tradeoff curves, we can find a more isometric representation space but with a loss in reconstruction accuracy (there is a tradeoff between the "correctness of the manifold" and "how isometric the latent space is").
> > > For example, different class images are collapsed in Subfigure I because the regularization coefficient was too big. Subfigure F seems to have a good balance.
> > >
> > > We completely agree that adding a simpler example will show more clearly whether our method is effective. In fact, during our past experiments, by using MNIST digits 0 and 1 only, we have observed that class separability was improved by a very large margin. Thank you for your nice suggestion. We will add this additional experimental result in the final version of the Appendix.

---

> ### Author Response · Authors · 2021-11-17
> **Response to Reviewer MiWx (1)**
>
> Thank you very much for your constructive feedback, and our sincerest
> apologies for not uploading our comments sooner. In response to the
> many constructive suggestions we have received, we have spent the
> past week conducting a wide range of new experiments, and revising
> our manuscript accordingly. In an attempt to answer the reviewer
> questions, and to better clarify and validate our contributions,
> the appendix has been substantially expanded with the following
> additional content:
>
> * Appendix B: Pseudocode;
> * Appendix C: Comparison to other regularization approaches that use
> 	the Jacobian;
> * Appendix E.1: Advantages of isometric regularization from a
> 	generative perspective;
> * Appendix E.3: Ablation study on mixup parameter;
> * Appendix E.4: Computational speed;
> * Appendix E.5: Isometric regularization for other autoencoders;
> * Appendix E.6: Isometry vs scaled isometry;
> * Appendix E.7: Multiple runs for computing standard deviations;
> * Appendix E.8: Sensitivity analysis of the latent space dimension.
>
> Below we provide detailed responses to each of the reviewer comments.
> When referencing any major changes and addition of new content to the
> revised manuscript, we have indicated those passages ``like this``.
>
> __Q1.__ I am not convinced that the method leads to any benefit on
> MNIST. The MSE gain compared to FMVAE is really marginal and could result
> from non-optimal hyperparameter tuning.
>
> __A1.__ We welcome your skepticism and the opportunity to further
> explain the details of our experiments.  In our case the most important
> hyperparameter that can affect training is the regularization coefficient
> $\alpha$, and we have taken great care to ensure that our comparison is
> made in an $\alpha$-invariant manner. Specifically, we compare the
> tradeoff curves, i.e., MSE as a function of VoR and MCN, obtained
> by using a range of regularization coefficients.  The remaining parameter,
> the mixup coefficient $\eta$, is set to $0.2$ in both algorithms
> for fair comparison ``(an ablation study
> on the mixup parameter has been added in Appendix E.3)``. We can assert
> with some confidence that the MSE gain is not the result of hyperparameter
> tuning. Rather, it is a consistently observed phenomenon, as we observe
> the same results over a range of different experimental settings, e.g.,
> in Figure 3 for CMU data and Figure 6 for diverse image data now
> included in Appendix E.2.
>
> In addition, following the suggestion of reviewer ZGHS, ``we
> have repeated the same experiment multiple times over different random
> seeds, and reported the averages and standard deviations in Appendix E.7.`` From these additional experiments, it has become even
> more evident that our algorithm outperforms the existing FMVAE algorithm.
> ``Beyond these empirical validations of our claim,
> we have also followed the suggestion of Reviewer 5XZF, and added a
> more detailed discussion in Appendix C of why our coordinate-invariant
> approach outperforms FMVAE.``
>
> __Q2.__ Figures 1 and 2, the MSE vs. MCN and MSE vs. VoR from
> both of these plots are redundant; you could only leave the ones in
> Figure 2; the information is redundant. What are the colors in the
> embedding? The labels? If so, the separation is not improved between
> classes.
>
> __A2.__ VoR and MCN measure two different aspects of the
> pullback Riemannian metric $J_f(z)^T J_f(z)$: VoR measures
> the degree of homogeneity of the Riemannian metric over the
> latent space (i.e., the extent to which the Riemannian metric
> varies with $z$), whereas MCN measures the degree of isotropy
> of the Riemannian metric at a given $z$ (that is, visualizing the
> Riemannian metric at $z$ as an ellipsoid, how close this ellipsoid
> is to being a circle).  Both measures should be considered
> in tandem when evaluating how close the mapping $f$ is to being
> a scaled isometry.
>
> The colors indeed represent labels.  Subfigure F of Figure 1 shows
> improved separation between classes compared to the subfigure A,
> which is expected.  Subfigure I of Figure 1 is a case of
> poor separation as a result of the regularization coefficient
> $\alpha$ being too large (implying a large reconstruction error).
> These figures clearly show the underlying tradeoff between
> reconstruction accuracy and the degree of isometry of the latent
> space, particularly the influence of the regularization
> coefficient $\alpha$.

---

### Official Review · Reviewer_ZGHS · 2021-11-02

**Correctness:** 3
**Technical Novelty And Significance:** 2
**Empirical Novelty And Significance:** 2
**Recommendation:** 5
**Confidence:** 4

**Main Review:**

**Strengths:**

I appreciate the theoretical motivation and foundation for the method proposed. Its justification and development were clear enough to follow. The experimental results in an unsupervised image retrieval task show a noticeable improvement in performance measured by multiple metrics when compared to a strong baseline. Furthermore, the tradeoffs between achieving a scaled isometry and balancing the reconstruction error are on the side of the proposed method, as evaluated on image and motion capture data.

**Weaknesses:**

While I mentioned most experimental results are convincing, I still think the paper has some considerable flaws to be addressed.

First, the experimental results do not seem averaged over multiple random seeds. Additional information such as standard deviation, would be helpful in assessing how stable the training is and how reliable are those results for someone attempting to reproduce the experiments. The same applies to the curves shown in Figures 1 and 2, where the difference to the baseline is marginal. I see this as critical for the paper to be accepted.

Second, there seems to be a conflict between the KL-divergence regularization of a VAE w.r.t. changes introduced to the latent space caused by the newly introduced regularization scheme. I believe there should be a discussion on this, since knowing $P_Z$ seems to be necessary and without the KL term it is hard to say it follows a standard Gaussian.

Third, I missed a discussion with other perspectives, such as from [1], where instead of changing the latent space that can be learned, one considers different ways of "navigating" it as is.

**Questions for the authors:**

- I found the first listed contribution to be problematic, which says the authors define a family of coordinate-invariant regularization terms. It strikes me as straightforward since it seems to be a by-product of the necessary mathematical foundation for the method. The family of functions itself was not really explored, just one specific case. Could you make it clearer how novel is that family of regularization terms w.r.t. existing work on this?
- Regarding the generative performance of the models in question. While I understand this might not be the main goal of the paper, the methods used as starting point (VAE) do present the characteristic of being able to generate new data points. For instance, because we change the structure of the latent space, how does this affect the generative aspect of the model?
- Why is it the case that IRVAE outperforms IRVAE+FM w.r.t. to P@1 by so much, but this does not happen in any other case?
- What are the actual impacts of the additional computational costs of the regularization procedure, in terms of total training time compared to not using them?

**Additional minor comments:**
- Section 2 seemed more like a subsection, as it only deals with introducing some concepts and relating them to each other, without carrying any major point on its own.
- There are some problems with notation, specifically the function h which seems to have multiple meanings (c.f. Eq. (6) and Eq. (11))
- It is also usual to also credit [2] for VAEs.
- Section 5.1: what is considered a sufficiently large number of samples for computing MCN and VoR?
Figures 1 and 2 show categorical information (classes of data points) using a continuous color scale. While this does not affect interpretation, I believe a categorical color scale would improve readability.
- References seem to need review. For instance, the first and fifth are from ICML 2020 and ICLR 2017, respectively, but this information is missing.

[1] Arvanitidis, Georgios, Lars Kai Hansen, and Søren Hauberg. "Latent Space Oddity: on the Curvature of Deep Generative Models." International Conference on Learning Representations. 2018.

[2] Rezende, Danilo, and Shakir Mohamed. "Variational inference with normalizing flows." International conference on machine learning. PMLR, 2015.


**Summary Of The Paper:**

The paper introduces a regularization scheme that constrains autoencoders to produce latent spaces which are isometrically closer to the input space, up to a certain scale factor, that is, a latent space whose pullback metric is closer to a scaled identity matrix. The paper further shows both (i) the tradeoff between two surrogate metrics for isometry and reconstruction accuracy, showing how much reconstruction error is affected by increasingly stricter regularization and (ii) that the representations learned by a variational autoencoder with such a regularizer results in better performance on a retrieval task when compared to existing baselines.

**Summary Of The Review:**

The paper follows a solid theoretical foundation using (scaled) isometry between input and latent space as a principle for designing a regularizer to learn meaningful representations with autoencoders. Experimental results show improvements over relevant baseline methods in an unsupervised retrieval task and as tradeoffs between reconstruction accuracy and surrogate metrics for how close the learned latent space is to an (scaled) isometry. These positive aspects, however, are overshadowed by key major issues: (i) results are not averaged over multiple random seeds, which makes it difficult to assess the stability of the regularizer in training; (ii) and missing discussion of conflicting requirements of the KL divergence term with the one introduced, making the method analysis seem too superficial.

---

> ### Author Response · Authors · 2021-11-17
> **Response to Reviewer ZGHS (2)**
>
>  __Q3.__ Third, I missed a discussion with other perspectives, such
> as from [1], where instead of changing the latent space that can be
> learned, one considers different ways of "navigating" it as is.
>
> __A3.__ We appreciate and agree with your comment, which was also
> raised by other reviewers. Our isometric decoder has several desirable
> features from the perspective of data generation as well (e.g., interpolation
> or latent value modulation). ``We have added in Appendix E.1
> results of additional experiments that demonstrate some of these
> other advantages.``
>
> __Q4.__ I found the first listed contribution to be problematic,
> which says the authors define a family of coordinate-invariant regularization
> terms. It strikes me as straightforward since it seems to be a by-product
> of the necessary mathematical foundation for the method. The family of
> functions itself was not really explored, just one specific case. Could
> you make it clearer how novel is that family of regularization terms
> w.r.t. existing work on this?
>
> __A4.__ The study of coordinate-invariant (or to be more accurate,
> approximately coordinate-invariant) distortion measures that
> measure how close the mapping is to being an isometry has a long
> history in the machine learning literature; these measures have been
> surveyed and discussed in, e.g., (Jang et al., 2020). As far as we
> know, we believe it's accurate to claim that we're the first to point
> out that using ``strict" isometric distortion measures (i.e., those
> that measure the nearness to a strict isometry in the sense of
> Equations (4) and (6)) can in fact be detrimental, and that a
> relaxed distortion measure, in the sense of Equations (5),(7),
> (8),(10) --  we believe it's the first time such measures have
> appeared in the ML literature -- is generally more effective.
>
> Returning to the question of why we did not further explore other choices,
> the fundamental reason can be traced to Proposition 2.  One can define any
> number of relaxed distortion measure by choosing an appropriate convex function
> $h$ and symmetric function $S$.  However, from a practical implementation
> perspective, the choice of $h$ and $S$ can have significant consequences.
> As we tried to explain in the paragraph immediately above Proposition 2,
> the computation of the relaxed distortion measure and its gradient (or
> back-propagation including the computation of the full Jacobian),
> requires considerable memory and computational resources. The specific
> choices made in Proposition 2 are with these practical considerations
> in mind, so that learning can be achieved within a reasonable time frame
> (by using the stochastic trace estimator).  The practical advantages
> offered by our method are an equally important component of the overall
> research significance of our contribution, we believe.
>
> __Q5.__ What are the actual impacts of the additional computational
> costs of the regularization procedure, in terms of total training time
> compared to not using them?
>
> __A5.__ We appreciate the question, which was also raised by other
> reviewers. ``We have added in Appendix E.4 results of
> additional experiments that compare computational speeds.``
>
> __Q6.__ There are some problems with notation, specifically the
> function h which seems to have multiple meanings (c.f. Eq. (6) and Eq. (11))
>
> __A6.__  Thank you for pointing this out; we have changed the $h$
> appearing in Sec. 4.2 to $i$.
>
> __Q7.__ It is also usual to also credit [2] for VAEs. References
> seem to need review. For instance, the first and fifth are from ICML 2020
> and ICLR 2017, respectively, but this information is missing.
>
> __A7.__ Thank you for pointing out these omissions and errors. We
> have revised the paper accordingly, adding the suggested reference and
> correcting errors.

---

> ### Author Response · Authors · 2021-11-17
> **Response to Reviewer ZGHS (1)**
>
> Thank you very much for your constructive feedback, and our sincerest
> apologies for not uploading our comments sooner. In response to the
> many constructive suggestions we have received, we have spent the
> past week conducting a wide range of new experiments, and revising
> our manuscript accordingly. In an attempt to answer the reviewer
> questions, and to better clarify and validate our contributions,
> the appendix has been substantially expanded with the following
> additional content:
> * Appendix B: Pseudocode;
> * Appendix C: Comparison to other regularization approaches that use
> 	the Jacobian;
> * Appendix E.1: Advantages of isometric regularization from a
> 	generative perspective;
> * Appendix E.3: Ablation study on mixup parameter;
> * Appendix E.4: Computational speed;
> * Appendix E.5: Isometric regularization for other autoencoders;
> * Appendix E.6: Isometry vs scaled isometry;
> * Appendix E.7: Multiple runs for computing standard deviations;
> * Appendix E.8: Sensitivity analysis of the latent space dimension.
>
> Below we provide detailed responses to each of the reviewer comments.
> When referencing any major changes and addition of new content to the
> revised manuscript, we have indicated those passages ``like this``.
>
> __Q1.__ First, the experimental results do not seem averaged over
> multiple random seeds. Additional information such as standard deviation,
> would be helpful in assessing how stable the training is and how reliable
> are those results for someone attempting to reproduce the experiments. The
> same applies to the curves shown in Figures 1 and 2, where the difference
> to the baseline is marginal. I see this as critical for the paper to be
> accepted.
>
> __A1.__ Thank you for the excellent suggestion, we completely
> agree with your recommendation. ``We have performed the same
> set of experiments multiple times over different random seeds, and now
> report the averages and standard deviations; the results are described
> in detail in Appendix E.7``.  The averaged tradeoff curves of IRVAE (our proposed method)
> are clearly located lower than those of FMVAE, with the standard
> deviations (visualized as ellipsoids) correspondingly small. We believe
> these results go even further to show that our algorithm IRVAE demonstrably
> outperforms FMVAE.  ``In addition to the additional
> empirical evidence supporting our claims, following the suggestion of
> Reviewer 5XZF, we have added in Appendix C a more detailed discussion of
> why our coordinate-invariant approach outperforms FMVAE.``
>
> __Q2.__ Second, there seems to be a conflict between the KL-divergence
> regularization of a VAE w.r.t. changes introduced to the latent space
> caused by the newly introduced regularization scheme. I believe there
> should be a discussion on this, since knowing $P_Z$ seems to be necessary
> and without the KL term it is hard to say it follows a standard Gaussian.
>
> __A2.__ While this may already be apparent to the reviewer, we'd
> first like to point out that that use of VAE and the KL term is not strictly
> necessary in order to apply our algorithm. Sampling from $P_Z$ proceeds as
> follows in our algorithm:
>
> * Sample data $x_i,x_j$ from the given data set;
> * Encode data $x_i,x_j$ to $z_i,z_j$ using the encoder $g_{\phi}$;
> * Augment the encoded data using mixup as needed, i.e., $z=\delta z_1 +
> (1-\delta) z_2$ where $\delta$ is uniformly sampled from $[-\eta, 1+\eta]$.
>
> We do not sample $z$ from the prior distribution (the standard Gaussian);
> instead, we sample $z$ from the encoded data distribution.
> ``To clarify this point, we have added the relevant pseudocode
> in Appendix B.``
>
> The use of VAE is not necessary as already mentioned; our method is
> applicable to other types of autoencoders. ``We have
> added additional experiments in Appendix E.5 that show the effects
> of our proposed regularization term when applied to other autoencoders``.
>
> Returning to the reviewer's first question about a possible conflict with
> the KL-divergence regularization term in VAE, precisely because of the
> scale-invariance property of our measure (please see the third condition
> in the second paragraph of Section 3.3), our geometric regularization
> term does not have any conflict with the KL term. The primary effect of
> the KL-divergence term (under the choice of a standard Gaussian) is to
> decrease the norm of the encoded latent values; one way to think of this
> is as the assignment of a prior on the latent space volume. Trying to force
> the decoder to be a strict isometry more often than not will conflict
> with this latent space volume prior. In contrast, our relaxed distortion
> measure attempts to find a scaled isometry that by design does not conflict
> with this volume prior; the relaxed distortion measure does not favor any
> particular volume of the latent space. ``We have further
> clarified these differences between seeking a strict isometry versus a
> scaled isometry in the newly added Appendix E.6.``

---

> > ### Comment · Reviewer_ZGHS · 2021-11-30
> > **Response to Authors**
> >
> > I appreciate the authors' reaction to my comments and criticism. They have addressed all major issues I have raised, most critically added more random seeds in their experiments and averaged the results, which maintained their claims. Their response to other reviewers is also appreciated.
> >
> > I would change my score to 6 (marginally above the acceptance threshold). The clarifications about the sampling procedure and the value of their contribution also played an important role in that.

---

> > > ### Author Response · Authors · 2021-11-30
> > > **Response to Reviewer ZGHS**
> > >
> > > Thank you for acknowledging our contribution and raising the score! Your comments have helped a lot to improve our paper. We sincerely appreciate this reviewing process!

---

### Official Review · Reviewer_5xzF · 2021-11-03

**Correctness:** 3
**Technical Novelty And Significance:** 3
**Empirical Novelty And Significance:** 3
**Recommendation:** 8
**Confidence:** 3

**Main Review:**

The paper is clear and well-motivated. The problem of introducing geometry properties to the latent space is challenging and not much explored. The proposed regularization method is convincing and well justified. The idea of coordinate invariant functional is novel (for the best of my knowledge) and theoretically sound.
The paper is well organized. The state of the art is complete. The description of method is clear. It would be nice to see a graphical scheme of the method architecture to give a visual explanation to the meaning of the involved functions and their role to move from one space to the other.
Experiments are convincing. The experiments on the most simpler scenarios like MNIST and CMU motion capture are useful to understand the behavior of the proposed approach. The experiment on the most challenging dataset, i.e., unsupervised human face retrieval is important to appreciate the improvement of the proposed approach against other methods.
There are some important points that authors should clarify:
-In the introduction, authors claimed that using a less stringent regularization term on isometry is more helpful, but than is not clear to identify other parts of the paper that support this claim. Please clarify with more examples and details.
-It is not clear the role of the stabilizing latent space flattening described in Sec 4.3. If this term is relevant it should be described in Eq. (11).
-Although it is already clear that the proposed method is better than FMVAE, authors should explain more details on that, especially in highlighting the importance of a coordinate-invariant strategy.
-h in Eq. (7) is different than h in Sec. 4.2, perhaps it would be better to change notation…,
-It would be nice to see the benefit of the proposed method for data generation. Usually to evaluated the regularity of the latent space,  the generation of data in between two samples is visually shown. Is it possible to show something similar to appreciate the benefit of the geometric constraints?
-Is the code available? More details on the used software and performance in terms of speed of method will be appreciated.


**Summary Of The Paper:**

The paper proposes a new regularization approach to introduce geometric constraint to the latent space of an autoencoder. A hierarchy for geometry-preserving mappings is formulated to clarify how strong this constraint can be defined. In particular, authors focused on scaled isometries, i.e., maps that preserve angles and distances up to some scale factor. This scale is learnt together with the manifold and the latent space representation during the training of the autoencoder. This approach was proposed also in (Chen et al., 2020) for the so called FMVAE, but here a new coordinate-invariant regularization term is introduced that measure how close the decoder
is to being a scaled isometry. Finally, a post-processing flattening procedure is introduced to further improve the geometry properties of the latent space.
Several results have been reported on different applicative scenarios showing a clear improvement of the proposed approach in comparison with other methods of the state of the art .


**Summary Of The Review:**


The proposed method is novel and convincing.
The paper is well organized and clear.
Experiments shown that the proposed approach clearly overcome other methods.

---

> ### Author Response · Authors · 2021-11-17
> **Response to Reviewer 5xzF (2)**
>
> __Q2.__ In the introduction, authors claimed that using a less stringent
> regularization term on isometry is more helpful, but then is not clear to
> identify other parts of the paper that support this claim. Please clarify
> with more examples and details.
>
> __A2.__ Thank you for pointing out this deficiency. ``We
> have now added additional experimental results in Appendix E.6 that support our
> claim that finding a scaled isometry is better than finding a strict isometry.``
>
> __Q3.__ It would be nice to see the benefit of the proposed method
> for data generation. Usually to evaluated the regularity of the latent
> space, the generation of data in between two samples is visually shown.
> Is it possible to show something similar to appreciate the benefit of the
> geometric constraints?
>
> __A3.__ Excellent suggestion, thank you. The isometric decoder indeed
> possesses several desirable features from the perspective of data
> generation (e.g., interpolation or latent value manipulation).
> ``We have added additional experimental results
> in Appendix E.1 that better show these advantages.``
>
> __Q4.__ It is not clear the role of the stabilizing latent space
> flattening described in Sec 4.3. If this term is relevant it should be
> described in Eq. (11).
>
> __A4.__ ``As suggested, we have moved this description
> to Eq. (11), and added an explanation of why the regularization is needed.``
>
> __Q5.__ h in Eq. (7) is different than h in Sec. 4.2, perhaps it
> would be better to change notation…,
>
> __A5.__ Thank you. We have changed the $h$ appearing in Sec. 4.2 to $i$.
>
> __Q6.__ Is the code available? More details on the used software
> and performance in terms of speed of method will be appreciated.
>
> __A6.__ Source code will be made available very soon (we are in
> the process of cleaning it up and adding documentation).
> ``We have added in Appendix E.4 results of additional
> experiments that compare computational speeds.``

---

> > ### Comment · Reviewer_5xzF · 2021-11-30
> > **Response to Authors**
> >
> > I am satisfied with the author's responses to my questions. I confirm my positive feeling on this work.

---

> > > ### Author Response · Authors · 2021-12-01
> > > **Response to Reviewer 5xzF**
> > >
> > > Thank you again for recognizing the value of our research. Your questions and comments have helped greatly in the revision of the paper!

---

> ### Author Response · Authors · 2021-11-17
> **Response to Reviewer 5xzF (1)**
>
> Thank you very much for your constructive feedback, and our sincerest
> apologies for not uploading our comments sooner. In response to the
> many constructive suggestions we have received, we have spent the
> past week conducting a wide range of new experiments, and revising
> our manuscript accordingly. In an attempt to answer the reviewer
> questions, and to better clarify and validate our contributions,
> the appendix has been substantially expanded with the following
> additional content:
>
> * Appendix B: Pseudocode;
> * Appendix C: Comparison to other regularization approaches that use
> 	the Jacobian;
> * Appendix E.1: Advantages of isometric regularization from a
> 	generative perspective;
> * Appendix E.3: Ablation study on mixup parameter;
> * Appendix E.4: Computational speed;
> * Appendix E.5: Isometric regularization for other autoencoders;
> * Appendix E.6: Isometry vs scaled isometry;
> * Appendix E.7: Multiple runs for computing standard deviations;
> * Appendix E.8: Sensitivity analysis of the latent space dimension.
>
> Below we provide detailed responses to each of the reviewer comments.
> When referencing any major changes and addition of new content to the
> revised manuscript, we have indicated those passages ``like this``.
>
> __Q1.__ Although it is already clear that the proposed method is
> better than FMVAE, authors should explain more details on that, especially
> in highlighting the importance of a coordinate-invariant strategy.
>
> __A1.__ We agree with the reviewer's observation that our paper could better highlight the importance of a coordinate-invariant strategy.
> Geometric objects that we want to preserve, such as length, angle, and volume, are all coordinate invariant concepts, which require a coordinate invariant formulation from the start.
> If one uses a coordinate-variant measure, then there is no guarantee that it will work effectively for different choices of coordinates.
>
> In this regard, we first show that the regularization term defined in FMVAE is not coordinate-invariant.
> Let $\mathcal{M}$ be a Riemannian
> manifold of dimension $m$ with local coordinates $z\in\mathbb{R}^{m}$
> and Riemannian metric $G(z) \in \mathbb{R}^{m \times m}$, and
> $\mathcal{N}$ be a Riemannian manifold of dimension $n$ with local
> coordinates $x \in \mathbb{R}^n$ and Riemannian metric $H(x) \in
> \mathbb{R}^{n \times n}$.  Let $\mathrm{f}:\mathcal{M} \to
> \mathcal{N}$ be a smooth mapping, represented in local coordinates
> by the italic symbol $f: \mathbb{R}^{m} \to \mathbb{R}^{n}$.
> Let $J_f$ be the Jacobian of $f$ and $P_Z$ be the probability distribution expressed in $\mathbb{R}^{m}$. Let $\nu$ be a positive measure on $\mathcal{M}$.
>
> The regularization term defined in FMVAE is then
> $$\||J_f^T(z)H(f(z)) J_f(z) - cG(z)\||^2_F,$$
> where $\||\cdot\||^2_F$ is the Frobenius norm and $c=\int_{\mathcal{M}}\frac{1}{m}
> \mathrm{Tr}(J_f^T(z) H(f(z)) J_f(z))d\nu$.
> For simplicity, consider a pair of linear coordinate transformations on the input manifold $z'=Az$ and output manifold $x'=Bx$. Then the function $f$ is transformed to $f'(z'):=B f(A^{-1}z')$, $G(z)$ is transformed to $G'(z'):=A^{-T}G(z)A^{-1}$ and $H(x)$ is transformed to $H'(x'):=B^{-T}H(x)B^{-1}$. Then, after some calculations, the above regularization term is transformed to
> $$\big\||A^{-T}\big(J_f^T(z)H(f(z)) J_f(z) - c'G(z)\big)A^{-1}\big\||^2_F,$$
> where $c'=\int_{\mathcal{M}}\frac{1}{m}
> \mathrm{Tr}(A^{-T}J_f^T(z) H(f(z)) J_f(z)A^{-1})d\nu$. Recall that the Frobenius norm $\||A\||_F^2=\frac{1}{2}\mathrm{Tr}(A^T A)$, the $A^{-T}$ and $A^{-1}$ multiplied at sides are not canceled; we can see that this is not coordinate-invariant.
>
> In addition to this, perhaps a more direct reason that the FMVAE does not
> perform as well as our method can be explained as follows.
> We remark that one of the desired properties of the scaled isometry
> measure is the third condition in Section 3.3: given two mappings $f$
> and $g$ from $\mathcal{M}$ to $\mathcal{N}$, if $J_f^T(z)
> H(f(z))J_f(z)=c J_g^T(z) H(g(z)) J_g(z)$ for some $c>0$ for all $z$,
> then the scaled isometry measure should be the same (that is, the
> metric should not distinguish between $f$ and $g$). This makes
> the measure more natural, in the sense that it does not a priori
> favor a particular scale for the pullback metric; if the pullback
> metrics are equivalent up to some scale, then these should be treated
> the same.
>
> The regularization term introduced in FMVAE can also be viewed as
> a scaled isometry measure. However, it fails to satisfy the
> the all-important third property.
> Even though the pullback metrics of $f$ and $g$ are equivalent up to some scalar multiplication, i.e., $J_f^T(z) H(f(z))J_f(z)=c J_g^T(z) H(g(z)) J_g(z)$ for some $c>0$ for all $z$, the one with the "smaller" Jacobian (i.e., smaller norm)
> achieves a lower value of the regularization term. As a result,
> the FMVAE favors a mapping with a smaller Jacobian, which can
> be detrimental to learning accurate data manifolds.
> ``We have now added this discussion in Appendix C.``

---

### Official Review · Reviewer_YhcR · 2021-11-06

**Correctness:** 3
**Technical Novelty And Significance:** 3
**Empirical Novelty And Significance:** 2
**Recommendation:** 5
**Confidence:** 3

**Main Review:**

Pros:

The idea of studying geoemetry-preserving regularization while learning representations is a challenging and an important topic with a large potential impact.

Cons:

1) In section 4.2 the paper talks about using a seperate invertible mapping from R^m -> R^m that essentially looks at not affecting the reconstruction accuracy and only improves the isometric representation. I am not sure why this can not be already captured by the original decoder that takes an input point in R^n-> R^m

2) The experiments show that the isometric regularization and FM leads to marginal or no loss in reconstruction accuracy. In the implementation H(x) is assumed to be identity and this is a strong assumption. As the authors acknowledge, it would be more useful to use domain-specific knowledge. However, by assuming identity for H(x) the regularization gets closer to Jacobian regularization that is already a well studied problem:

  Rifai, S., Dauphin, Y. N., Vincent, P., Bengio, Y., and Muller, X. The manifold tangent classifier. In Advances in Neural Information Processing Systems, pp. 2294–2302, 2011a.

   Rifai, S., Mesnil, G., Vincent, P., Muller, X., Bengio, Y., Dauphin, Y., and Glorot, X. Higher order contractive auto-encoder. In
ECML-PKDD, pp. 645–660. Springer, 2011b.

   Hoffman et al. Robust Learning with Jacobian Regularization, 2019.

3) The visualization in Fig. 1 does not clearly show the improvement using the isometric regularization approach shown in the paper.

4) There should be some ablation on the mixup regularization that is used in this paper. It is not clear whether the improvement is mainly from mixup or the proposed regularization.

5)  The paper does not look at sufficient strong baselines for representational learning methods. It only uses VAE and FMVAE.

**Summary Of The Paper:**

This paper studies the aspect of preserving geometry on the learned latent space representations. In particular, the paper looks at a hierarchy of geometry-preserving mapping (isometry, conformal mapping of degree k, area preserving mapping, etc.). Existing popular methods such as SimCLR pays limited attention to preserving geometric relationships. The paper shows that the mapping that preserves angles and relative distances is better than the ones that preserve angles and absolute distances. The main contribution of this paper is to propose a representational learning technique that uses a reconstruction loss and an isometric regularization term. This allows us to learn embeddings that satisfy isometric properties while not suffering from any reconstruction loss or just marginal reconstruction loss. The comparison is done with VAE (Kingma & Welling, 2014) and FMVAE (Chen et al. 2020). Experiments are shown on CelebA with 40 annotations.

**Summary Of The Review:**

Overall, the paper addresses an important problem but there are many concerns with the experiments and the formulation.

---

> ### Author Response · Authors · 2021-11-17
> **Response to Reviewer YhcR (2)**
>
>
> __Q3.__ There should be some ablation on the mixup regularization that
> is used in this paper. It is not clear whether the improvement is mainly
> from mixup or the proposed regularization.
>
> __A3.__ We should first point out that in our context, mixup alone
> cannot be used without our proposed regularization term. Unlike the
> conventional mixup augmentation technique performed in the data space, the
> mixup augmentation in this paper is introduced for the purpose of computing
> the geometric regularization term, and is done in the latent space.
> ``To better clarify this point, we have added relevant pseudocode
> in Appendix B.`` ``Still, we believe the reviewer's suggestion
> that some ablation is warranted is valid, and in Appendix E.3 we have
> performed ablation studies on the mixup regularization parameter.``
>
> __Q4.__ In Section 4.2 the paper talks about using a separate
> invertible mapping from $R^m \to R^m$ that essentially looks at not
> affecting the reconstruction accuracy and only improves the isometric
> representation. I am not sure why this cannot be already captured by
> the original decoder.
>
> __A4.__
> In an ideal scenario where (i) there is infinite training data available and
> (ii) the decoder is expressive enough to represent any nonlinear function,
> the first stage of our isometric regularization procedure should be enough
> to produce a sufficiently isometric representation space. However, in practice
> this is very rarely the case. Although the second condition can be made almost
> true by using a sufficiently flexible neural network, due to the lack of
> training data, the isometric regularization procedure is not enough to produce
> a sufficiently isometric representation space.
>
> To explain why, we first need to describe how learning actually proceeds.
> Note that in isometric regularization, the loss function consists of two
> terms: (i) a reconstruction error term, and (ii) a geometric regularization
> term.  As is typical when training models in general machine learning
> contexts, we minimize the loss on the training set, and stop training before
> the onset of overfitting.  In our case, the onset of overfitting is determined
> by monitoring the reconstruction error of the validation set (good
> reconstruction is a priority, after all).  If the reconstruction error
> for the validation set increases repeatedly over some fixed number
> of iterations, training is stopped. As a result, while there may
> still be room to lower the geometric regularization term even further,
> autoencoder learning can stop.  If we had infinite data, then there
> would be no need to monitor for the onset of overfitting, but
> since this is not the case, we instead rely on the flattening module.
>
> __Q5.__ The paper does not look at sufficient strong baselines for
> representational learning methods. It only uses VAE and FMVAE.
>
> __A5.__ Our main focus in this paper is to show the effects of the
> proposed regularization term in the context of unsupervised representation
> learning with autoencoders.
> Therefore, we believe other recent self-supervised learning approaches such
> as SimCLR are not appropriate targets for comparison.  Among the countless
> variants of autoencoders, we choose one of the most popular variants, the
> VAE, as the baseline for comparison. We should also emphasize that
> our method is indeed applicable to many other types of autoencoders.
> ``We have therefore added in Appendix E.5 additional
> experiments that show the effects of the proposed regularization term
> for more diverse autoencoders.``

---

> > ### Comment · Reviewer_YhcR · 2021-11-29
> > **Response to authors**
> >
> > I really appreciate the response on the comparison with Jacobian regularization and mixup experiments.

---

> > > ### Author Response · Authors · 2021-11-30
> > > **Response to Reviewer YhcR**
> > >
> > > Thank you for recognizing our efforts. If you have any further questions or comments, please feel free to ask.

---

> ### Author Response · Authors · 2021-11-17
> **Response to Reviewer YhcR (1)**
>
> Thank you very much for your constructive feedback, and our sincerest
> apologies for not uploading our comments sooner. In response to the
> many constructive suggestions we have received, we have spent the
> past week conducting a wide range of new experiments, and revising
> our manuscript accordingly. In an attempt to answer the reviewer
> questions, and to better clarify and validate our contributions,
> the appendix has been substantially expanded with the following
> additional content:
>
> * Appendix B: Pseudocode;
> * Appendix C: Comparison to other regularization approaches that use
> 	the Jacobian;
> * Appendix E.1: Advantages of isometric regularization from a
> 	generative perspective;
> * Appendix E.3: Ablation study on mixup parameter;
> * Appendix E.4: Computational speed;
> * Appendix E.5: Isometric regularization for other autoencoders;
> * Appendix E.6: Isometry vs scaled isometry;
> * Appendix E.7: Multiple runs for computing standard deviations;
> * Appendix E.8: Sensitivity analysis of the latent space dimension.
>
> Below we provide detailed responses to each of the reviewer comments.
> When referencing any major changes and addition of new content to the
> revised manuscript, we have indicated those passages ``like this``.
>
> __Q1.__ In the implementation $H(x)$ is assumed to be identity and this is a strong assumption. As the authors acknowledge, it would be more useful to use domain-specific knowledge. However, by assuming identity for $H(x)$ the regularization gets closer to Jacobian regularization that is already a well studied problem: (i) the manifold tangent classifier and (ii) higher order contractive auto-encoder.
>
> __A1.__ Even with $H(x)=I$, our isometric regularization term differs
> significantly from existing approaches that regularize the Jacobian norm
> or its higher-order derivatives.  The key difference is that while existing
> Jacobian norm regularization approaches attempt to seek *smooth mappings*
> that are close to being linear, our isometrically regularized autoencoders go
> further and attempt to find *geometry-preserving mappings*, i.e., mappings
> in which distances, angles, and volumes are preserved between the latent space
> and input data space. To be more precise, denoting the Jacobian of $f(x)$ by
> $J_f(x)$, the objective of our regularization term is to make
> $$J_f(x)^T J_f(x) = cI$$
> for some $c>0$;  any $f$ satisfying the above is a *\it scaled isometry*.
> Our goal is not to seek $f$ that minimizes $\| J_f(x)\|$ (note that $f =$
> constant results in a zero-norm Jacobian); naturally the requirement
> that $f$ reconstruct the data ensures that $f$ cannot be constant, but
> using the Jacobian norm as the regularization term essentially seeks an
> approximate linear map that best reconstructs the data. This is quite
> different from our aim of seeking an $f$ that preserves the gemoetry.
> In fact, our regularization term is designed in a scale-invariant way so
> that the measure does not depend on the scale of the Jacobian (please
> refer to the third condition in the second paragraph of Section 3.3).
> ``Your comment has convinced us that we need to better
> clarify these differences, and in Appendix C we have now included an
> in-depth discussion on the differences between our regularization term
> and other existing regularization approaches based on the Jacobian``.
>
> __Q2.__ The visualization in Fig. 1 does not clearly show the
> improvement using the isometric regularization approach shown in the paper.
>
> __A2.__ The MNIST experiments in Figures 1 and 2 are intended to
> show the following:
>
> * There is an inherent tradeoff between the reconstruction error
> and how isometric the latent space is. Specifically, (i) the tradeoff
> curves are obtained by varying the regularization coefficients $\alpha$,
> and (ii) sub-figures A, B, F, I with corresponding equidistance plots
> illustrate these tradeoffs (F, I produce a more isometric latent space
> but less accurate reconstruction, while the opposite holds for A, B);
> * the IRVAE learns more isometric representations than existing
> methods (FMVAE) *at any level of reconstruction accuracy*.
>
> Figure 1 is intended to illustrate these two points.
>
> In addition, following the suggestion of reviewer ZGHS, ``we
> have repeated the same experiment multiple times over different random
> seeds, and reported the averages and standard deviations (Appendix E.7).`` From these additional experiments, it has become even
> more evident that our algorithm outperforms the existing FMVAE algorithm.
>
> To address any questions about the latent space dimension used in the
> experiments, we intentionally use two-dimensional latent spaces to
> visualize the equidistance plots. Similar experiments with more diverse
> image data (MNIST, FMNIST, SVHN, CIFAR10) and higher-dimensional latent
> spaces have been added in Appendix E.2.

---

### Decision · Program_Chairs · 2022-01-20

**Decision:**

Accept (Poster)

**Comment:**

This paper proposes an extension to learning a representation: it motivates, proposes and evaluates a new regularizer term that promotes smoothness via enforcing the representation to be geometry-preserving (isometry, conformal mapping of degree k). Comparisons with a standard VAE and FMVAE (Chen et al. 2020) are shown and experiments are provided on CelebA with several different attributes as target classification tasks.

The paper has received extensive reviews and the authors have successfully answered most of the concerns raised, mostly regarding comparisons to other techniques that try to introduce a regularization based on the properties of the Jacobian of the decoder network.
The appendix has been extended as a result of the rebuttal and the paper could be accepted.

Notes:
I find the formulation based on the notion of the isometric decoder somewhat surprising as the encoder is a key object of interest that controls the nature of the representation. The authors should clarify the assumption 3 in 3.3 better by the consideration of potentially $dim(z) << dim(x)$, how the isometry of the decoder effects the encoder,

Additionally, for the latent space flattening an ablation using SVD (merely a linear mapping for $i(\cdot)$) could be considered.

Reviewer ZGHS has noted that they raise their grade to 6 in their comment, but this is still not currently reflected.